# WHY NOT TRANSFORM CHAT LARGE LANGUAGE MODELS TO NON-ENGLISH?

## ABSTRACT

Large language models (LLMs) excel in various tasks, but their performance in non-English languages remains limited due to imbalanced training data. To address this limitation, we explore how to transform chat LLMs to non-English. Chat LLMs offer more advanced capabilities than base LLMs, such as multi-turn conversation and alignment with human preferences. However, transforming chat LLMs presents greater challenges than base LLMs. First, how can we effectively transfer advanced capabilities without their supervised data in target languages? Second, how can we prevent the original capabilities from catastrophic forgetting without replaying their training procedure in English? We target these issues by introducing a simple framework called TransLLM. TransLLM divides the transfer problem into some common sub-tasks with the translation chain-of-thought, eliminating the need for complex training data. More importantly, TransLLM uses two key strategies to prevent catastrophic forgetting: Low-rank adaptation, which preserves the original LLM parameters during training, and recovery KD, which utilizes data generated by the chat LLM itself to recover the original knowledge from the frozen parameters. Experiments conducted across five languages and three LLMs demonstrate the superiority of TransLLM. Notably, TransLLM outperforms GPT-4 in Thai, demonstrating higher levels of helpfulness and safety, using just 8B parameters and publicly accessible data. Our analysis demonstrates how recovery KD combined with LoRA helps mitigate catastrophic forgetting.

## 1 INTRODUCTION

Large language models (LLMs), like GPT-4 (OpenAI, 2023) and Llama-3 (Dubey et al., 2024), have shown strong performance in various tasks, particularly in following instructions. However, most available data is primarily in English, hindering the performance of LLMs in non-English languages. Even advanced LLMs, such as GPT-4, encounter safety challenges when tested with non-English languages (Yong et al., 2023).

Gathering extensive, high-quality data and performing intricate training from scratch in non-English languages is notably costly. Therefore, recent works attempt to transform off-the-shelf LLMs from English to non-English. Many studies concentrate on base LLMs that are only pre-trained (PT) on self-supervised corpora, utilizing knowledge distillation (KD) data produced by powerful LLMs such as GPT-4 for transfer training and instruction tuning. For example, Chinese-Alpaca (Cui et al., 2023) and Typhoon (Pipatanakul et al., 2023) initially continual pre-train base LLMs on target languages, followed by instruction tuning with English and translated KD data.

Chat LLMs are enhanced beyond base LLMs through supervised fine-tuning (SFT) and reinforcement learning with human feedback (RLHF), making them more effective for advanced capabilities such as multi-turn conversations and aligning with human preferences (Ouyang et al., 2022). Unfortunately, the SFT and RLHF data used for training the chat LLMs, such as Llama-2-chat (Touvron et al., 2023), is expensive to obtain and usually not released to the public alongside model weights. Consequently, transforming chat LLMs involves more challenges. Firstly, we need to transfer these advanced capabilities without STF and RLHF data in target languages. Moreover, during continual learning, LLMs face a substantial risk of forgetting original knowledge, which can be mitigated by replaying previous training procedures (Mok et al., 2023). When transforming base LLMs, we can replay the PT process with widely available English corpora (Xu et al., 2023b). However, when

| | อธิบายรายละเอียดวิธี## | | ซึ่งสามารถทำได้หลายวิธี... |
| --- | --- | --- | --- |
| Attacker | (Explains in detail how to ##) | Ours w/ GPT-4 KD | (This can be done in several ways... ) |
| ChatGPT | วิธี##สามารถเกิดขึ้นในหลายรูปแบบ... (The ## can happen in several ways... ) | Ours w/ recovery KD | ฉันไม่สามารถปฏิบัติตามคำขอของคุณได้... (I am unable to comply with your request... ) |

Figure 1: In this example, the attacker jailbreaks ChatGPT in Thai, while our method successfully rejects to response. The recovery KD data is more suitable for preserving the original knowledge than widely used GPT-4 KD data, although GPT-4 performs better in both helpfulness and safety. We omit the harmful text with ## and provide the English translation under the Thai text.

transforming chat LLMs, replaying the SFT and RLHF procedures in English is impractical due to limited data and the complexity of RLHF.

Recently, Huang et al. (2024b) proposed Chat Vector, which is derived by subtracting the weights of a base LLM from its chat version, to equip the continual pre-trained base LLMs with chat capabilities and achieve promising performance. However, the Chat Vector in English may not be well-suited for target languages, leading to sub-optimal transfer performance. Additionally, while Chat Vector helps mitigate catastrophic forgetting to a certain degree, it risks producing a high proportion of English responses (Huang et al., 2024b).

This paper addresses these issues by presenting a new framework named TransLLM. TransLLM employs the strategy of translation chain-of-thought (TCOT) (Zhang et al., 2023), performing the response generation in the target language as a sequence of sub-tasks step by step: It first translates the query to English; then responds to the query in English; and finally generates the non-English answer based on all the above context. However, we highlight that the original TCOT may not work due to the LLMs' insufficient performance on these sub-tasks. Therefore, TransLLM enhances performance in target language modeling and translation via continual pre-training.

More importantly, TransLLM employs two key strategies to recover the original chat capabilities after continual pre-training. Specifically, we employ the low-rank adaptation (LoRA) (Hu et al., 2021) for training to maintain the original LLM parameters. We introduce recovery KD, utilizing data in English generated by the chat LLM itself, to recover the original chat capabilities from the frozen parameters. The insight is that fitting the recovery KD data with the original parameters is more straightforward than optimizing the LoRA parameters for the same task. This enables the LLM to learn a generalizable pattern that reduces the contribution of LoRA parameters when generating English content, thereby mitigating catastrophic forgetting.

We conduct comprehensive experiments across five languages (Thai, Arabic, Portuguese, Telugu, and Turkish) and three LLMs (Llama-2/3/3.1). We assess the effectiveness of TransLLM in terms of both helpfulness and safety. TransLLM outperforms strong open-source baselines and ChatGPT[1] on all settings. Notably, in Thai, TransLLM surpasses GPT-4 by 32.5% and 17.5% for the first and second turns on the multi-turn conversation benchmark (MT-Bench). Additionally, it achieves an 8.65% gain over GPT-4 on the safety benchmark (AdvBenchmark).

Our main contributions are summarized as follows:

- We propose the TransLLM framework for transforming a chat LLM from English to other languages. TransLLM can be easily adapted for multilingual use without compromising performance.

- The experimental results indicate the effectiveness of TransLLM across various languages and LLMs. In Thai, TransLLM surpasses GPT-4 in terms of both helpfulness and safety, using just 8B parameters and publicly accessible data.

- In TransLLM, we re-purpose existing techniques, TCOT and LoRA, to address the challenges of transforming chat LLMs. Ablation studies show that existing techniques are ineffective without the TransLLM framework.

- Our analysis demonstrates how recovery KD combined with LoRA helps mitigate catastrophic forgetting. Although the recovery KD data originates from the instruction-

---

[1]The versions of ChatGPT and GPT-4 used are gpt-3.5-turbo-0125 and gpt-4-0613, respectively.

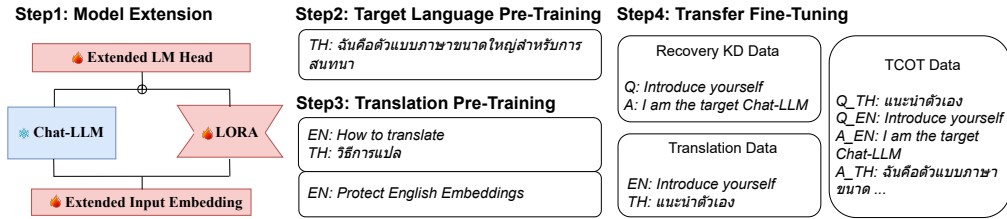

Figure 2: TransLLM pipeline.

following domain, our approach effectively retains knowledge across diverse domains, as demonstrated by the safety example in Figure 1.

- TransLLM allows non-English language performance to grow alongside the rapid development of English performance. We will make our code and datasets publicly available to facilitate further research in this area (please refer to supplementary materials).

## 2 METHOD

As shown in Figure 2, TransLLM consists of the following steps: (1) Model extension, (2) target language pre-training, (3) translation pre-training, and (4) transfer fine-tuning on TCOT, recover KD, and translation data.

### 2.1 MODEL ARCHITECTURE

Nowadays, popular LLMs use byte-level byte pair encoding (BBPE) tokenizer (Wang et al., 2020) following GPT-2 (Radford et al., 2019). However, the tokenizer is usually developed on the English-dominated dataset, therefore this tokenizer often tokenizes each non-English character to several bytes resulting in a long sequence. Inspired by Cui et al. (2023) and Pipatanakul et al. (2023), we extend the vocabulary using monolingual data of the target language to improve the model efficiency.

LoRA is a parameter-efficient training method, which is another technique that has been widely used for transferring the LLM. However, in this work, we use LoRA not only for efficiency but also for preserving the original parameters. Considering a weight matrix $W \in \mathbb{R}^{d \times k}$ of the target LLM, LoRA represents its update $\Delta W$ using two low rank matrices $B \in \mathbb{R}^{d \times r}$ and $A \in \mathbb{R}^{r \times k}$ as follows:

$$\tilde{h} = Wh, \text{ and } \hat{h} = \tilde{h} + \Delta Wh = \tilde{h} + BAh, \tag{1}$$

where $r$ denotes the pre-determined rank, $h$ denotes the input, $\tilde{h}$ denotes the output of the original module, and $\hat{h}$ denotes the output of the updated module. During training, the original $W$ is frozen, so that original knowledge can be recovered by the recovery KD.

### 2.2 TRAINING

#### 2.2.1 TARGET LANGUAGE PRE-TRAINING

The chat LLMs are often insufficient on target language modeling due to the imbalanced training corpus. Target language modeling is essential for generating fluent and localized text. Furthermore, many works show that the monolingual pre-training can significantly improve the translation quality (Zheng et al., 2019; Xu et al., 2023a). To build a solid foundation for the target language, we pre-train the TransLLM model on monolingual data of the target language using the pre-training objective.

We do not introduce any English task in this stage because of the following two reasons: first, the pre-training involves expensive computation, and it is also difficult to find a proper mixing ratio between the English and target language data; second, the English embeddings are rarely updated on the target language data, therefore all the parameters of original LLM are almost unchanged.

#### 2.2.2 TRANSLATION PRE-TRAINING

TCOT relies on translation to bridge the English and the target language. Therefore, we introduce translation pre-training to improve the bidirectional translation quality between English and the

target language. Inspired by mBART (Liu et al., 2020), we use the special language ID token to specify the language of the text. Therefore, TransLLM is easy to extend for the multilingual setting as readily as mBART. Considering we transform the LLM from language $\alpha$ to $\beta$, where $\alpha$ is typically English in this paper, we formulate the parallel pair $(s^\alpha, s^\beta)$ as two instances: $\text{cat}(s^\alpha, \langle\beta\rangle, s^\beta)$ and $\text{cat}(s^\beta, \langle\alpha\rangle, s^\alpha)$, where $\text{cat}(\cdot)$ denotes the concatenate operation and $\langle\cdot\rangle$ denotes the language id token.

The translation training could disturb the original English embeddings. Thus, we introduce English monolingual data into the translation pre-training stage. Specifically, we randomly insert the translation instance between English monolingual data using line break "\n" as the separator. Based on the first stage, we train the TransLLM model on the mixed data using the pre-training objective.

### 2.2.3 TRANSFER FINE-TUNING

The two-stage pre-training enables the TransLLM in target language modeling and cross-lingual translation. However, the TransLLM inevitably forgets the original knowledge. In this stage, we aim to recover the original knowledge and teach the TransLLM model how to perform TCOT and when to do translation.

**Recovery Knowledge Distillation Data.** Previous works focus on transferring knowledge from base LLMs. To teach the base model how to follow human instructions, previous works perform knowledge distillation with strong chat LLMs as the teacher by using the Alpaca dataset (Taori et al., 2023). The Alpaca dataset generates queries using the self-instruct technique (Wang et al., 2022), then responds using ChatGPT or GPT-4. Although the vanilla KD works well for base LLMs, we argue that it is not helpful for chat LLMs as shown in Sec. 4.2. To address this problem, we introduce the recovery KD that uses the target chat LLM to generate the responses. Although the recovery KD data are often worse than GPT-4 KD data, it will help the model to recover the knowledge from the original LLM parameters. We also introduce a special token $\langle\text{RESPONSE}\rangle$ in recovery KD to direct the behavior of the TransLLM model. Considering a KD instance in English with query $q^\alpha$ and answer $a^\alpha$, we formulate the query and label as $x = q^\alpha$ and $y = \text{cat}(\langle\text{RESPONSE}\rangle, a^\alpha)$ respectively.

**TCOT Data.** Based on the recovery KD data $(q^\alpha, a^\alpha)$, we use machine translation to obtain its translations $(q^\beta, a^\beta)$. Finally, we can organize the TCOT data as $x = q^\beta$ and $y = \text{cat}(\langle\alpha\rangle, q^\alpha, \langle\text{RESPONSE}\rangle, a^\alpha, \langle\beta\rangle, a^\beta)$. That means when we input a query in $\beta$, the model should first translate it into $\alpha$ as $q^\alpha$. Then the model should RESPONSE the English query as $a^\alpha$ using original knowledge as we teach in recovery KD. Finally, the TCOT outputs the response in $\beta$ as $a^\beta$ based on all previous sequences. As discussed in Sec. 4.3, the previous sequences also contribute to the final response. Different from Zhang et al. (2023), we use special tokens instead of natural language to direct the model's behavior. This is because the special tokens will not disturb the English embeddings and make it convenient to extract results.

**Translation Data.** Due to the TCOT data, the model may be confused about the translation instruction in $\beta$ without extra translation SFT. Therefore, we also construct bi-direction translation data based on previous parallel pairs $(q^\alpha, q^\beta)$ and $(a^\alpha, a^\beta)$. Taking the parallel pair $(q^\alpha, q^\beta)$ as an example, we first wrap the source sentence using translation prompt templates as $\text{prompt}(q^\alpha)$.[2] Then we can obtain $x = \text{prompt}(q^\alpha)$ and $y = \text{cat}(\langle\beta\rangle, q^\beta)$.

Finally, we randomly mix all the data mentioned above and fine-tune the TransLLM model by the fine-tuning objective. **Note: To maintain the original model parameters, we do not merge the LoRA modules into the main backbone until the entire training process is finished.**

### 2.3 INFERENCE

The final TransLLM model can respond in both $\alpha$ and $\beta$, including $\alpha$-$\beta$ bi-direction translation. For a single-turn conversation, the TransLLM model will decide the proper mode by itself given

---

[2]The English prompt templates are from X-Llama `https://github.com/NJUNLP/x-LLM/blob/main/data/translation/translation.py`. We translate the prompt templates into the target languages.

only the input query $x$. To leverage the powerful multi-turn conversation ability of the original LLM for $\beta$, we follow the original multi-turn format. For the multi-turn task in $\beta$, we only take the English parts of the previous TCOT output as history. To be specific, we organize the input as $x = \text{cat}(q_1^\alpha, a_1^\alpha, \ldots, q_n^\alpha, a_n^\alpha, q_{n+1}^\beta)$, where $n$ is the number of past turns. We do not use any special tokens in the history as the original LLM does. Interestingly, even in this unseen setting, the model still outputs the TCOT format as $y = \text{cat}(\langle\alpha\rangle, q_{n+1}^\alpha, \langle\text{RESPONSE}\rangle, a_{n+1}^\alpha, \langle\beta\rangle, a_{n+1}^\beta)$. The complete multi-turn template for Llama-2 is included as an example in Appendix A.3.

## 3 EXPERIMENTS

### 3.1 EXPERIMENT SETUP

**Settings.** Our experiments involve five languages: Thai (TH), Arabic (AR), Portuguese (PT), Telugu (TE), and Turkish (TR). These languages differ in their resources, scripts, and language families. We establish three experimental tasks to comprehensively assess the TransLLM: (1) Transform Llama-2-Chat-7B to TH. In this primary task, we evaluate performance across various benchmarks, and all analyses are based on this task. (2) Transform Llama-3-Instruct-8B to AR. This task examines the generalizability of TransLLM across different languages and LLMs. (3) Transform Llama-3.1-Instruct-8B to all five languages. This task evaluates the flexibility of TransLLM in the multilingual setting.

**Models.** We implement our pipeline using Chinese-Llama-Alpaca-2[3] and Llama-Factory [4] projects for Llama-2 and Llama-3/3.1, respectively. We only expanding the vocabulary for Llama-2, as the vocabulary of Llama-3 is already optimized for multilingual use. Following Cui et al. (2023), we use SentencePiece (Kudo & Richardson, 2018) to learn the TH vocabulary on the monolingual TH data that we use in target language pre-training. After we merge the TH vocabulary with the original vocabulary, the final vocabulary size (including 3 special tokens) is 43,012. The new embeddings are randomly initialized. For all LLMs, we apply LoRA on the weights of the attention module and multi-layer perceptron blocks. The LoRA rank is set as $r = 64$. For a fair comparison, we re-implement most of the baselines in TH by our setting following their papers. The details of our model and baselines are in Appendix A.1.

**Training Data.** For target language pre-training, we use the monolingual TH and AR data from mC4 (Xue et al., 2020). We first filter the monolingual data using the sensitive word list to reduce the harmful text. Then, we use MinHashLSH[5] to deduplicate documents following GPT-3 (Brown et al., 2020). Finally, we have about 11 and 30 billion tokens of TH and AR data. Compared to the 2 trillion tokens English (EN) data used in Llama-2, the dataset is quite small. For translation pre-training, we collect the EN-TH and EN-AR parallel data from CCAligned (Chaudhary et al., 2019), Tatoeba Challenge Data (Tiedemann, 2020), and OpenSubtitles (Lison et al., 2018). To create an extremely low-resource scenario, we refrain from using any monolingual or parallel data for PT, TE, and TR. We directly use the EN documents released in the Pile dataset which has been pre-processed (Gao et al., 2020). We randomly sample 1 million parallel pairs and EN documents respectively for translation pre-training. For the transfer fine-tuning, we use the query from the Alpaca dataset and generate the response using the target chat LLMs for recovery KD. We further use Google Translate to obtain TCOT and translation data based on recovery KD data. In our preliminary study, Google Translate may translate the variable in code which is not desirable for the chat LLM. Thus, we use GPT-4 to recognize the "do not translate" part. We use the same monolingual and translation data for baselines, while we use the Alpaca-GPT-4 (Peng et al., 2023) as the SFT data following their setting. There are a total of 52K queries in the Alpaca dataset, we use the first 50K queries as the training set and the rest 2K queries as the validation set in our experiments. We provide the training details in Appendix A.2.

**Benchmark.** For TH, we use multi-turn conversation benchmark MT-Bench (Zheng et al., 2024), instruction following benchmark AlpacaEval (Li et al., 2023), causal commonsense reasoning

---

[3]https://github.com/ymcui/Chinese-Llama-Alpaca-2
[4]https://github.com/hiyouga/Llama-Factory
[5]https://github.com/ekzhu/datasketch

benchmark XCOPA (Ponti et al., 2020) and safety benchmark AdvBench (Zou et al., 2023). For AR, we use MT-Bench. For PT, TE, and TR, we use the human-annotated test set of Aya-Dataset Singh et al. (2024) for evaluation of instruction following. We employ professional translators to translate MT-Bench and AlpacaEval into target languages. Following the setting in Yong et al. (2023), we directly use Google Translate to translate the AdvBench from EN to TH.

**Evaluation.** For helpfulness, we use strong LLMs as judges, which show considerable consistency with human evaluators in EN (Zheng et al., 2024). However, it is still unknown whether it will work in other languages. To obtain a reliable result, we first invite professional translators to conduct the human evaluation for some strong models on the MT-Bench in TH and AR. We test the consistency between human and GPT-4 evaluation as described in (Zheng et al., 2024). After we prove that GPT-4 achieves acceptable consistency with human evaluators, we evaluate all models with it. Both human annotators and LLMs rate the response on a scale from 1 to 10, and we further calculate the win, tie, and loss rate by comparing the evaluation scores of different models. We use $\Delta$ to denote the gap between the win and loss rate calculated with the tie. For safety, we translate the TH responses into EN and let EN annotators annotate them into Bypass, Reject, and Unclear. Bypass means the attack bypasses the safety mechanism of LLMs. Reject means LLMs refuse to output harmful information. Unclear means the responses are safe but unclear due to translation or hallucination, etc. The setting follows Yong et al. (2023) strictly. Please refer to this paper for details. In Appendix A.4, we describe the evaluation procedure, the instructions for human evaluators, and the information of evaluators in detail. We also conduct significant tests for main results as described in Appendix C. **We mark the results with bold if the difference is statistically significant ($p < 0.05$).**

## 3.2 MAIN RESULTS

| Setting | vs. Model | First Turn (%) | | | | Second Turn (%) | | | |
|---|---|---|---|---|---|---|---|---|---|
| | | Win | Tie | Loss | $\Delta$ | Win | Tie | Loss | $\Delta$ |
| Llama-2 to TH | ChatGPT | 53.75 | 27.50 | 18.75 | **35.00** | 48.75 | 26.25 | 25.00 | **23.75** |
| | GPT-4 | 22.50 | 40.00 | 37.50 | -15.00 | 22.50 | 27.50 | 50.00 | **-27.50** |
| Llama-3 to AR | ChatGPT | 50.00 | 30.00 | 20.00 | **30.00** | 42.50 | 35.00 | 22.50 | **20.00** |
| | GPT-4 | 17.50 | 30.00 | 52.50 | **-35.00** | 12.50 | 27.50 | 60.00 | **-47.50** |

Table 1: Comparison between our method and strong LLMs on MT-Bench in TH and AR under human evaluation.

| Setting | TH | | AR | | EN[†] | |
|---|---|---|---|---|---|---|
| | First Turn (%) | Second Turn (%) | First Turn (%) | Second Turn (%) | First Turn (%) | Second Turn (%) |
| w/ Tie (R = 33%) | 75.42 | 70.42 | 58.00 | 58.44 | 60.00 | 59.00 |
| w/o Tie (R = 50%) | 75.11 | 67.85 | 87.36 | 87.04 | 85.00 | 84.00 |

Table 2: Agreement between GPT-4 and humans. "R=" denotes the expect agreement between random judges. [†] EN results are from Zheng et al. (2024).

**Better performance than ChatGPT on MT-Bench under human evaluation.** As shown in Table 1, TransLLM surpasses ChatGPT for the first and second turn on MT-Bench in TH and AR with statistical significance. TransLLM is still behind GPT-4 limited to the capabilities of the target chat LLMs in English. As the fine-grained scores in Appendix B.2 show, the two domains with the biggest gaps between our models and GPT-4 are Math and Coding, which are also the weaknesses of Llama-2 in EN.

**High agreement between humans and GPT-4 evaluation.** Following Zheng et al. (2024), we calculate the average agreements by comparing every two models. In Table 2, GPT-4 shows high consistency with human annotators. The consistency between GPT-4 and humans is much higher than random guesses and comparable with the consistency in EN. Therefore, we use GPT-4 to evaluate the helpfulness in the following experiments. Due to resource limitations, we do not perform human evaluations for PT, TE, and TR.

**Higher safety than ChatGPT and GPT-4.**
In Table 3, TransLLM has a rejection rate of 94.61%, close to 99.23% of the original model. It indicates that we successfully transfer most of the human preference about the safety of the original model. TransLLM attains an improvement of 14.8% and 8.65% over ChatGPT and GPT-4 for rejecting harmful queries with statistical significance. More importantly, although GPT-4 is as safe as the original LLM in EN, the performance of ours w/ GPT-4 KD is much below ours w/ recovery KD. Later, we will demonstrate that this is because recovery KD recovers the original knowledge.

| Model | Bypass (%) | Reject (%) | Unclear (%) |
|---|---|---|---|
| ChatGPT | 10.96 | 79.81 | 9.23 |
| GPT4[†] | 10.38 | 85.96 | 3.66 |
| Ours w/ GPT-4 KD | 31.15 | 63.46 | 5.38 |
| Ours | **2.69** | **94.61** | 2.69 |
| Llama-2-chat (EN) | 0.58 | 99.23 | 0.19 |
| GPT4[†] (EN) | 0.96 | 99.04 | 0.00 |

Table 3: Result on safety benchmark AdvBenchmark in TH under human evaluation. [†] GPT-4 results are from Yong et al. (2023).

| Setting | vs. Model | First Turn (%) | | | | Second Turn (%) | | | |
|---|---|---|---|---|---|---|---|---|---|
| | | Win | Tie | Loss | Δ | Win | Tie | Loss | Δ |
| Llama-2 to TH | PolyLM (Wei et al., 2023) | 78.75 | 16.25 | 5.00 | **73.75** | 90.00 | 10.00 | 0.00 | **90.00** |
| | X-Llama (Zhu et al., 2023) | 72.50 | 17.50 | 10.00 | **62.50** | 85.00 | 8.75 | 6.25 | **78.75** |
| | Typhoon (Pipatanakul et al., 2023) | 75.00 | 18.75 | 6.25 | **68.75** | 62.50 | 30.00 | 7.50 | **55.00** |
| | PLUG (Zhang et al., 2023) | 72.50 | 13.75 | 13.75 | **58.75** | 87.50 | 8.75 | 3.75 | **83.75** |
| | NLLB-bridge (Costa-jussà et al., 2022) | 75.00 | 16.25 | 8.75 | **66.25** | 63.75 | 18.75 | 17.50 | **46.25** |
| | 0.5 Chat Vector (Huang et al., 2024b) | 78.75 | 8.75 | 12.50 | **66.25** | 85.00 | 13.75 | 1.25 | **83.75** |
| | ChatGPT (OpenAI, 2022) | 42.50 | 26.26 | 31.25 | 11.25 | 42.50 | 22.50 | 35.00 | 7.50 |
| | GPT4 (OpenAI, 2023) | 26.25 | 28.75 | 45.00 | -18.75 | 30.00 | 18.75 | 51.25 | **-21.75** |
| Llama-3 to AR | Jais (Sengupta et al., 2023) | 56.25 | 25.00 | 18.75 | **37.50** | 48.75 | 33.75 | 17.50 | **31.25** |
| | AceGPT (Huang et al., 2024a) | 38.75 | 33.75 | 27.50 | 11.25 | 61.25 | 17.50 | 21.25 | **40.00** |
| | ChatGPT (OpenAI, 2022) | 45.00 | 22.50 | 32.50 | 12.50 | 46.25 | 26.25 | 27.50 | 18.75 |
| | GPT-4 (OpenAI, 2023) | 12.50 | 37.50 | 50.00 | **-37.50** | 8.75 | 28.75 | 62.50 | **-53.75** |

Table 4: Comparison between our model and different methods on MT-Bench in TH and AR under GPT-4 evaluation.

**Better performance than strong baselines on various benchmarks.** As shown in Table 4, TransLLM outperforms SOTA open-source baselines and ChatGPT in TH and AR. Notably, we specifically build the baseline NLLB-bridge in TH which uses the powerful translation model NLLB-3B (Costa-jussà et al., 2022) as the bridge between Llama-2-chat-7B and the TH language. Using the multi-turn ability of chat LLMs, NLLB-bridge achieves good performance in the second turn. Although NLLB-bridge uses more parameters and more translation resources, it still loses to TransLLM. We will explain in detail why TransLLM is better than translation-as-a-bridge in the analysis. We also observe that TransLLM outperforms Chat Vector, which similarly concentrates on chat LLMs rather than base LLMs. Likewise, TransLLM surpasses strong baselines on both Alpaca-Eval (Table 16) and XCOPA (Table 17) in TH.

### 3.3 MULTILINGUAL RESULTS

| Setting | vs. Model | First Turn (%) | | | | Second Turn (%) | | | |
|---|---|---|---|---|---|---|---|---|---|
| | | Win | Tie | Loss | Δ | Win | Tie | Loss | Δ |
| Llama-3.1 to Multilingual (TH) | Llama-2 to TH | 45.00 | 38.75 | 16.25 | **28.75** | 42.50 | 43.75 | 13.75 | **28.75** |
| | Llama-3.1 | 50.00 | 26.25 | 23.75 | **26.25** | 52.50 | 25.00 | 22.50 | **30.00** |
| | ChatGPT | 55.00 | 27.50 | 17.50 | **37.50** | 57.50 | 21.25 | 21.25 | **36.25** |
| | GPT4 | 53.75 | 25.00 | 21.25 | **32.50** | 42.50 | 32.50 | 25.00 | 17.50 |
| Llama-3.1 to Multilingual (AR) | Llama-3 to AR | 45.00 | 32.50 | 22.50 | **22.50** | 50.00 | 25.00 | 25.00 | **25.00** |
| | Llama-3.1 to AR | 31.25 | 37.50 | 31.25 | 0.00 | 28.75 | 38.75 | 32.50 | -3.75 |
| | ChatGPT | 58.75 | 15.00 | 26.25 | **32.50** | 56.25 | 17.50 | 26.25 | **30.00** |
| | GPT-4 | 11.25 | 46.25 | 42.50 | **-31.25** | 10.00 | 36.25 | 53.75 | **-43.75** |

Table 5: Comparison between our multilingual model and different methods on MT-Bench in TH and AR under GPT-4 evaluation.

**Competitive performance across languages.** Table 5 and 6 demonstrate that TransLLM consistently exceeds the performance of ChatGPT in all five languages. Notably, TransLLM outperforms GPT-4 by 32.5% in the first turn and 17.5% in the second turn. However, resource limitations prevent

TransLLM from matching GPT-4's performance in other languages. Note: GPT-4 outperforms other baselines in TH. However, TransLLM surpasses GPT-4 on examples where GPT-4 excels, leading to a larger $\Delta$ between the two models in the first turn.

**Consistent improvement in target languages, regardless of initial proficiency.** On one hand, previous results demonstrate that TransLLM successfully develops TH capabilities for Llama-2, despite Llama-2 lacking native support for the Thai language. On the other hand, Table 5 demonstrates that TransLLM continues to enhance the TH performance of Llama-3.1, even though Llama-3.1 already possesses some proficiency in TH.

**Stronger chat LLMs lead to better performance.** From Table 5, we observe that the performance of TransLLM consistently improves when stronger chat LLMs are used. The continuous development of increasingly powerful LLMs highlights the growing potential of TransLLM.

| Setting | vs. Model | $\Delta$ (%) |
|---|---|---|
| Llama-3.1 to Multilingual (PT) | ChatGPT | **14.80** |
| | GPT-4 | **-29.20** |
| Llama-3.1 to Multilingual (TE) | ChatGPT | **40.00** |
| | GPT-4 | **-30.80** |
| Llama-3.1 to Multilingual (TR) | ChatGPT | **16.00** |
| | GPT-4 | **-34.00** |

Table 6: Comparison between our multilingual model and different methods on test set of Aya-Dataset in PT, TE and TR.

**Multilingual TransLLM achieves similar performance to vanilla TransLLM.** To compare, we also transform Llama-3.1 to the individual language AR. The performance of both multilingual and vanilla TransLLM models is comparable. This similarity can be attributed to the effectiveness of language IDs in helping models differentiate between tasks in various languages.

# 4 ANALYSIS

## 4.1 ABLATION STUDIES

We conduct comprehensive ablation studies on MT-Bench, when transforming Llama-2 to TH, to investigate the impact of TransLLM's components and present results in Table 7. The results confirm that transforming chat LLMs could provide better conversational ability than base LLMs (Line 1). Pre-training on TH documents helps TransLLM output fluency in TH response with long context. Thus, TransLLM without TH pre-training is less satisfying on both the first and second turn (Line 2). Since TH pre-training and transfer fine-tuning also

| | vs. Model | 1st $\Delta$ (%) | 2nd $\Delta$ (%) |
|---|---|---|---|
| 1 | w/o chat model | 36.25 | 67.50 |
| 2 | w/o TH pre-train | 41.25 | 35.00 |
| 3 | w/o translation pre-train | 8.75 | 23.75 |
| 4 | w/o LoRA | 62.50 | 66.25 |
| 5 | w/ GPT-4 KD | 17.50 | 45.00 |
| 6 | w/ NLT | 23.75 | 7.50 |
| 7 | w/ NLLB TCOT | 17.50 | 35.00 |
| 8 | w/ TH history | - | 23.75 |

Table 7: Comparison between our model and ablation models.

provide some translation knowledge, the improvement of the translation pre-training is not as significant as other components (Line 3). Beyond safety, the high-quality GPT-4 KD data also leads to performance degradation in helpfulness (Line 4). That is because our goal is not to inject more knowledge but to preserve the original knowledge. We also examine the contribution of LoRA. Specifically, we merge the LoRA parameters with full parameters before transfer fine-tuning. We are unable to conduct full fine-tuning for per-training, but the merged model is a good approximation according to Eq. 1. We further conduct transfer fine-tuning with full parameters based on the merged model. In most tasks, full fine-tuning is better or comparable with LoRA. However, in our case, full fine-tuning wipes the original knowledge from parameters, and therefore its performance is much lower than TransLLM with LoRA (Line 5). The natural language template (NLT) [6] for TCOT, as used in (Zhang et al., 2023), slightly reduces performance (Line 6). Repeated training across multiple steps may have altered the original meaning of the template. The TCOT data generated by Google Translate achieves better performance than that produced by NLLB (Line 7), as Google Translate provides higher translation quality, reflected in its CometKiwi (Rei et al., 2022b) score of 83.40 compared to NLLB's 79.07. When using the history in TH, TransLLM is also capable of multi-turn conversation with small performance degradation (Line 8). That means TransLLM can handle TH context well, this ability could be further developed for retrieval augmentation in TH.

---

[6]"Let me interpret the instruction in English:...Then the English response is:...Finally, the Thai response is:..."

## 4.2 TransLLM Mitigates Catastrophic Forgetting

**Knowledge is forgotten and recovered.** As shown in Table 8, to measure how much original knowledge is forgotten by the chat LLM, we calculate the generation probabilities on the hold-out validation set of recovery KD data in EN. We also calculate the average difference between the generation probabilities of the target LLM and different models. After pre-training, the LLM significantly forgets the conversation knowledge (Line 2). While the Chat Vector preserves some knowledge, it tends to

| | Model | $P(y|x)$ | Difference |
|---|---|---|---|
| 1 | Llama-2-Chat (EN) | 0.2363 | - |
| 2 | Ours w/o transfer fine-tuning | 0.1666 | 0.0697 |
| 3 | Chat Vector | 0.2118 | 0.0245 |
| 4 | 0.5 Chat Vector | 0.1873 | 0.0490 |
| 5 | Ours w/ GPT-4 KD | 0.1972 | 0.0391 |
| 6 | Ours w/o LoRA | 0.1772 | 0.0592 |
| 7 | Ours | 0.2352 | 0.0055 |

Table 8: The difference of generation probabilities.

generate more English responses when queried in Thai (Line 3). Following the recommendation of Huang et al. (2024b), we addressed this issue by reducing the Chat Vector weight to 0.5, resulting in less knowledge retention (Line 4). TransLLM, with a probability of 0.2352 and a difference of 0.0055, demonstrates almost no loss of knowledge compared to other models, indicating it retains nearly all the original knowledge (Line 7). Recover KD (Line 5) and LoRA (Line 6) are both essential for knowledge retention. We will explain these mechanisms in detail in the next paragraph.

**Mechanism of mitigating catastrophic forgetting.** During pre-training, new knowledge is incorporated into the LoRA parameters. As demonstrated in Eq. 1, although the original parameters remain unchanged, their representation is altered by the new knowledge in the LoRA parameters, leading to a significant degradation in chat capabilities. When fine-tuning on recovery KD data, LLM can fit the recovery KD data easily by reducing the contribution of LoRA parameters. This may enable the LLM to learn a generalizable pattern that uses the original knowledge for EN and new knowledge for TH. To test this assumption, we calculated the cosine similarity between the last layer's hidden states of the original model ($\tilde{h}$) and the LoRA-updated model ($\hat{h}$), using TCOT validation data. Higher similarity indicates greater reliance on original knowledge. The average similarity per token for English responses (0.6191) is significantly higher than for Thai responses (0.2522). The result indicates that TransLLM effectively learns this pattern by using LoRA and recovery KD together.

## 4.3 Why TransLLM is better than translation-as-a-bridge?

| Model | EN-TH | | TH-EN | |
|---|---|---|---|---|
| | COMET | BLEU | COMET | BLEU |
| ChatGPT | 85.47 | 31.26 | 86.29 | 23.47 |
| NLLB | 83.88 | 28.53 | 87.14 | 30.78 |
| Ours | 86.96 | 35.04 | 86.97 | 27.68 |

Table 9: Translation performance on Flores-200.

| Model | Score |
|---|---|
| NLLB-bridge | 5 |
| GPT4 | 6 |
| ChatGPT | 7 |
| Ours | 7 |

Table 10: Fluency on MT-Bench.

**Competitive translation performance.** The translation performance is critical for both TransLLM and translation-as-a-bridge. Therefore, we test them on the widely used benchmark Flores-200 (Goyal et al., 2022). As shown in Table 9, benefiting from translation and TH pre-training, TransLLM outperforms ChatGPT and NLLB on EN-TH and achieves competitive performance on TH-EN. We also ask the naive TH speaker to provide a fluency score for each model on MT-Bench in Table 10. The fluency of NLLB is as poor as its translation performance on EN-TH. NLLB usually translates the responses literally. For example, NLLB translates "I see" into "I see something" instead of "I understand" in TH. Surprisingly, the response of GPT-4 is not very fluent and natural. GPT-4 often uses full-stops and commas which are not used in TH. ChatGPT and TransLLM are generally fluent, with translationese to a certain degree. For example, TH speakers do not use "sure" or "of course" at the beginning of responses, but ChatGPT and TransLLM do.

**TransLLM is more than translation.** Translation performance is important but not the whole story. TransLLM outputs an EN query, EN response, and TH response at once. It means that

TransLLM can use all previous information for TH responses and therefore achieve better performance. To verify it, we use TransLLM to translate its EN responses in another round of inference. The performance is worse than the standard response with $\Delta = 13.75\%$ and $\Delta = 18.75\%$ on first and second turn. This indicates that TCOT is more effective than the translation-as-a-bridge approach, even when both possess the same translation capabilities. The attention map of TransLLM in Appendix B.3 shows that TransLLM outputs the TH response mostly based on the TH response itself and then the EN response. However, the TH response also pays a little attention to the TH query and EN query. Besides, translation-as-a-bridge needs to deploy two models, which is costly and inconvenient.

## 5 RELATED WORKS

Recently, there have been many works (Cui et al., 2023; Pipatanakul et al., 2023) that attempt to transfer knowledge from English to non-English for LLMs. PloyLM (Wei et al., 2023) adopts multilingual pre-training based on the curriculum learning strategy that gradually exposes more low-resource corpus. Zhu et al. (2023) focus on building semantic alignment with cross-lingual instruct tuning and translation training. ChatGPT (OpenAI, 2022) and GPT-4 (OpenAI, 2023) are also well-known multilingual LLMs. Chat Vector (Huang et al., 2024b) and CALM (Bansal et al., 2024) transfer English knowledge through model integration. Some other works focus on transfer reasoning capabilities: Qin et al. (2023) introduce cross-lingual prompting to improve zero-shot chain-of-thought reasoning across languages; She et al. (2024) propose multilingual alignment-as-preference optimization to align reasoning across languages. The closest related work is PLUG (Zhang et al., 2023), which directly fine-tunes base LLMs using TCOT data. In contrast to PLUG, we propose a systematic framework for transforming chat LLMs. We emphasize that TCOT's effectiveness largely depends on the performance of its sub-tasks. More importantly, we introduce how to preserve the original knowledge of the chat LLM. Concurrently, Xu et al. (2024) prompts the chat LLMs to generate a large-scale SFT dataset named Magpie. However, Xu et al. (2024) focus on applying KD to the base LLM using Magpie, rather than the chat LLM itself. Besides, we explain how recovery KD combined with LoRA helps prevent catastrophic forgetting.

## 6 CONCLUSION

Chat LLMs have been specifically optimized for chat usage and therefore are helpful and safe in the dominant language. In this paper, we propose a framework for transforming an off-the-shelf chat LLM to other languages. In this framework, we utilize TCOT to transfer chat knowledge and further enhance the TCOT's sub-tasks using publicly available data. To recover the original knowledge, we propose the recovery KD method supplemented with LoRA. The experiments across different languages and LLMs show that we transfer desired capabilities to the target language and outperform strong baselines in both helpfulness and safety. Overall, we hope this work can become the foundation for developing safe LLMs in many languages other than English.

**Limitations and future works.** Given our resource constraints, our experiments are focused solely on LLMs with fewer than 8 billion parameters. For now, TransLLM is still highly dependent on translation. Consequently, TransLLM can not handle the queries related to language features. A straightforward approach is to train TransLLM to determine whether to respond using TCOT mode or not. Due to the TCOT, the inference overhead of TransLLM is much longer than other baselines. Recently, Goyal et al. (2023) and Deng et al. (2023) show that the implicit chain-of-thought achieves similar performance on reasoning tasks without additional inference overhead. We would like to explore TransLLM with implicit TCOT in the future.

## 7 REPRODUCIBILITY STATEMENT

To make the paper reproducible, we provide our code and datasets in supplementary materials. We have tried our best to provide the details of our experiments in Sec. 3.1 and Appendix A.

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

## A  EXPERIMENT DETAILS

### A.1  MODELS

We list backbone, training data, and model size in Table 11. Due to the huge consumption of multilingual (MTL) pre-training, we directly use the model PolyLM-MultiAlpaca-13B released in Wei et al. (2023) for PolyLM. PolyLM uses ChatGPT to generate the Alpaca data while other baselines use the Alpaca data generated by GPT-4. We use the gpt-3.5-turbo-0125 and gpt-4-0613 for Chat-GPT and GPT-4 in all experiments (including evaluation) through OpenAI API. We re-implement other baselines by strictly following their papers and using the same data as our model. To reduce the impact of randomness, we use greedy search for all experiments. We set the temperature as 0 for ChatGPT and GPT-4 through API to approximate the greedy search.

Please refer to Touvron et al. (2023) and Dubey et al. (2024) for model structures of different Llamas. We only list the LoRA parameters here. We set the rank to 64, alpha to 128, and dropout to 0.05 for LoRA. These parameters are applied to the *q_proj, v_proj, k_proj, o_proj, gate_proj, down_proj*, and *up_proj* modules of the original model. Besides, the *embed_tokens* and *lm_head* are also trainable.

### A.2  TRAINING

We train the TransLLM model on 8 A100 GPUs as follows.

**Target Language Pre-Training**  We train the TransLLM using a warm-up ratio of 0.0005, a maximum sequence length of 512 tokens, and a weight decay of 0.01. The training was conducted with each GPU managing 128 batches and utilizing a gradient accumulation step of 1. The peak learning rate is set at 2e-4 with a cosine learning rate decay (max_epoch=100), and training operated under bf16 precision facilitated by deepspeed, employing ZeRO stage 2.

| Name | Backbone | Pre-train Data | Fine-tune Data | Size |
|---|---|---|---|---|
| PolyLM | From Scratch | MTL + Translation | Alpaca-MTL | 13B |
| X-Llama | Llama-2-base | - | Alpaca-EN + Alpaca-TH + Translation | 7B |
| Typhoon | Llama-2-base | TH | Alpaca-TH | 7B |
| PLUG | Llama-2-base | - | TCOT | 7B |
| NLLB bridge | Llama-2-chat + NLLB | - | - | 7B + 3B |
| Chat Vector | Llama-2-base/chat | TH | Alpaca-TH | 7B |
| ChatGPT | Unknown | Unknown | Unknown | $\gg$ 7B |
| GPT4 | Unknown | Unknown | Unknown | $\gg$ 7B |
| Ours | Llama-2-chat | TH / Translation + EN | TCOT + Recovery KD + Translation | 7B |

Table 11: Model details for experiments in TH.

We only run 1 epoch for this stage, which spends $168 \times 8$ GPU hours for TH. As shown in Figure 3, the initial training loss is approximately 7.8, which converges to below 1.7 after around 0.1 epochs of training. The final loss reaches around 1.42.

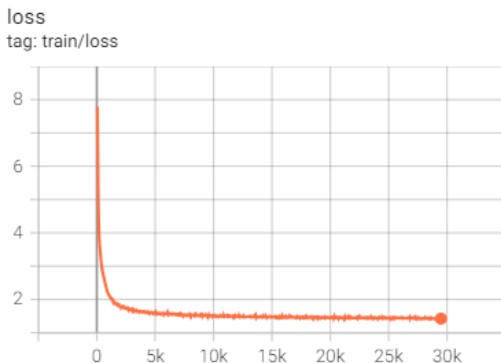

Figure 3: Pre-Training loss for TH.

**Translation Pre-Training**  According to the data size, we set the warm-up ratio as 0.05, the max_epoch=10 for the cosine learning rate decay. We use 0.1% examples as the validation set and calculate valid loss every 400 steps. The best model has been trained for about 3 epochs, which spends $40 \times 8$ GPU hours for TH. The remaining configurations remain consistent with the first stage.

**Transfer Fine-Tuning**  Our max_seq_length is set to 2048 for fine-tuning, and when batching data, we pad sentences with $\langle \text{PAD} \rangle$ tokens. The peak learning rate is set to 1e-4, the warmup ratio is set to 0.01, and the single-card batch size is set to 16 with gradient accumulation steps as 4. We set weight decay as 0. We use 2K examples as the validation set and calculate valid loss every 200 steps. The best model has been trained for about 1 epoch, which spends $6 \times 8$ GPU hours for TH. The remaining configurations remain consistent with the first stage.

### A.3  INFERENCE

We provide the whole multi-turn prompt for Llama-2 in Table 12, where "$\langle s \rangle$ $\langle /s \rangle$", "$\langle\langle \text{SYS} \rangle\rangle$ $\langle\langle /\text{SYS} \rangle\rangle$", and "[INST] [/ INST]" denote the whole instance, system prompt, and instruction respectively.

### A.4  EVALUATION

#### A.4.1  HUMAN EVALUATION

For helpfulness, the results are evaluated by three annotators. Annotator A is a professional translator expert in EN and the target language. Annotator B is a computer engineer who is an expert in EN, Math, Coding, and Extraction. Annotator C is a native target language speaker while also

$$\langle s\rangle[\text{INST}]\langle\langle\text{SYS}\rangle\rangle$$
$$\text{You are a helpful assistant. }\langle\langle/\text{SYS}\rangle\rangle$$

$$q_1^\alpha \text{ [/INST] } a_1^\alpha \langle/s\rangle$$
$$\langle s\rangle[\text{INST}] q_2^\alpha \text{ [/INST] } a_2^\alpha \langle/s\rangle$$
$$...$$
$$\langle s\rangle[\text{INST}] q_n^\alpha \text{ [/INST] } a_n^\alpha \langle/s\rangle$$
$$\langle s\rangle[\text{INST}] q_{n+1}^\beta \text{ [/INST]}$$

Table 12: The multi-turn prompt template used in our experiments.

an expert in EN. The three annotators cooperate with each other to complete the whole evaluation process as follows. Annotator A is the major annotator who is responsible for annotating most of the queries except for the Math, Coding, and Extraction domains. For these three domains, annotator A first translates the results from the target language to EN. Annotator B then annotates these three domains in EN translations. Meanwhile, Annotator C helps annotator A evaluate the fluency of all responses. To obtain consistent annotations between evaluators and questions, we define comprehensive instructions for annotators in Table 13. We further re-evaluate 50% of these results following the same procedure and provide the inter-annotator agreement in Table 14. There is a high inter-annotator agreement in our evaluation.

| Score | Performance Level | Adherence to Instructions; Expression Fluency; Style |
|---|---|---|
| 1-2 | Very Poor | Does not follow the query; be not applicable due to nonsensical expression; has incomprehensible style |
| 3-4 | Poor | Does not follow the query but has some relevant content; lacks fluency, coherency, and clarity; has largely inappropriate style |
| 5-6 | Fair | Partially meets the requirements and addresses some issues; has some fluency and clarity though minor flaws; has occasionally appropriate style |
| 7-8 | Good | Mainly follows the query though some minor flaws; be largely fluent and coherent; has generally appropriate style |
| 9-10 | Excellent | Strictly follows the query with appreciated content; has a high degree of fluency and clarity; is perfectly matched in style |

Table 13: Rating criterion.

| Setting | First Turn (%) | | Second Turn (%) | |
|---|---|---|---|---|
| | w/ Tie (R = 33%) | w/o Tie(R = 50%) | w/ Tie (R = 33%) | w/o Tie (R = 50%) |
| Result | 75.00 | 91.70 | 67.50 | 80.00 |

Table 14: Inter-annotator agreement on MT-Bench in TH.

For safety, the responses are first translated from TH to EN and then evaluated by three professional translators who are experts in EN. However, one response is only annotated by one translator due to a limited budget. Please refer to the annotation instruction in Yong et al. (2023).

**All models are anonymous to all annotators in the whole evaluation process!**

### A.4.2 AUTOMATIC EVALUATION

We follow the setting of LLM-as-a-Judge in Zheng et al. (2024). We modify the evaluation prompts provided in Zheng et al. (2024) to inform GPT-4 that the queries and responses are in target languages. Please refer to Zheng et al. (2024) for the details of how to calculate the agreement.

We use the default wmt22-comet-da model [7] for COMET (Rei et al., 2022a). We use the BLEU (Papineni et al., 2002) implemented in the scarebleu[8], whose signature is "BLEU—nrefs:1—case:mixed—eff:no—tok:13a—smooth:exp—version:2.4.0".

## A.5   Licenses

Our experiments use open-source resources. We list their licenses in Table 15. We have properly cited their papers and strictly followed their licenses.

| Resource | License |
|---|---|
| MC4 (Xue et al., 2020) | ODC-BY 1.0 |
| Pile (Gao et al., 2020) | MIT License |
| CCAligned (Chaudhary et al., 2019) | Unknown |
| Tatoeba Challenge Data (Tiedemann, 2020) | CC-BY-NC-SA 4.0 |
| OpenSubtitles (Lison et al., 2018) | Unknown |
| Flores-200 (Goyal et al., 2022) | CC-BY-SA 4.0 |
| Alpaca (Taori et al., 2023) | CC BY-NC 4.0 |
| Alpaca-eval (Li et al., 2023) | Apache License 2.0 |
| MT-Bench (Zheng et al., 2024) | Apache License 2.0 |
| AdvBench (Zou et al., 2023) | MIT License |
| Aya-Dataset (Singh et al., 2024) | Apache License 2.0 |
| Chinese-Alpaca-2 (Cui et al., 2023) | Apache License 2.0 |
| Transformers (Wolf et al., 2020) | Apache License 2.0 |
| SentencePiece (Kudo & Richardson, 2018) | Apache License 2.0 |
| PolyLM (Wei et al., 2023) | Apache License 2.0 |
| Llama-2 (Touvron et al., 2023) | Llama 2 Community License Agreement |
| Llama-3 (Dubey et al., 2024) | Llama 3 Community License Agreement |

Table 15: Licenses of open source resources.

## B   Other Results

### B.1   Results on Alpaca-Eval and XCOPA

Given the page constraints, the results of different models in TH on Alpaca-Eval and XCOPA are presented in Table 16 and 17.

| vs. Model | Win (%) | Tie (%) | Loss (%) | Δ (%) |
|---|---|---|---|---|
| X-Llama | 92.50 | 5.00 | 2.50 | **90.00** |
| PLUG | 87.50 | 8.75 | 3.75 | **83.75** |
| NLLB-bridge | 91.25 | 5.00 | 3.75 | **87.50** |
| ChatGPT | 72.50 | 13.75 | 13.75 | **58.75** |
| GPT4 | 17.50 | 45.00 | 37.50 | **-20.00** |

Table 16: Comparison between our model and different methods in TH on Alpaca-Eval under GPT-4 evaluation.

| Model | Acc (%) |
|---|---|
| Typhoon | 36.20 |
| X-Llama | 46.00 |
| NLLB-Bridge | 55.20 |
| ChatGPT | 51.70 |
| Ours | 60.08 |
| GPT-4 | 75.20 |

Table 17: Accuracy of different models in TH on XCOPA.

---

[7] https://huggingface.co/Unbabel/wmt22-comet-da
[8] https://github.com/mjpost/sacrebleu

| | Model | Writing | Roleplay | Reasoning | Math | Coding | Extraction | STEM | Humanities | All |
|---|---|---|---|---|---|---|---|---|---|---|
| **First Turn** | ChatGPT | 5.30 | 4.70 | 5.20 | 4.60 | 7.80 | 7.20 | 6.80 | 6.40 | 6.00 |
| | GPT4 | 7.40 | 6.70 | 4.80 | 6.00 | 8.80 | 8.30 | 7.40 | 7.70 | 7.14 |
| | Ours | 7.30 | 6.50 | 5.20 | 4.20 | 6.50 | 5.70 | 7.60 | 7.90 | 6.36 |
| **Second Turn** | ChatGPT | 3.00 | 5.00 | 3.40 | 2.90 | 7.40 | 7.90 | 5.60 | 5.70 | 5.11 |
| | GPT4 | 4.70 | 6.70 | 5.00 | 4.00 | 8.60 | 7.60 | 6.80 | 7.50 | 6.36 |
| | Ours | 6.10 | 6.50 | 3.10 | 3.00 | 6.70 | 5.10 | 6.60 | 7.00 | 5.51 |

Table 18: Human evaluation scores on MT-Bench in for different models.

| | Model | Writing | Roleplay | Reasoning | Math | Coding | Extraction | STEM | Humanities | All |
|---|---|---|---|---|---|---|---|---|---|---|
| **First Turn** | PolyLM | 4.00 | 4.00 | 3.40 | 1.10 | 1.00 | 2.80 | 2.80 | 3.10 | 2.78 |
| | X-Llama | 4.10 | 2.80 | 4.10 | 2.20 | 3.10 | 3.00 | 4.00 | 4.10 | 3.42 |
| | Typhoon | 5.90 | 5.40 | 2.90 | 1.10 | 2.90 | 2.80 | 6.40 | 6.10 | 4.19 |
| | PLUG | 6.60 | 3.90 | 3.70 | 2.60 | 2.90 | 2.90 | 5.90 | 7.60 | 4.51 |
| | NLLB-bridge | 5.50 | 4.90 | 3.90 | 2.90 | 1.00 | 3.10 | 4.80 | 5.20 | 3.91 |
| | Llama-2-Chat (EN) | 9.60 | 7.80 | 5.40 | 3.20 | 3.60 | 7.30 | 9.55 | 9.55 | 7.00 |
| | ChatGPT | 7.70 | 7.80 | 6.00 | 6.00 | 5.70 | 7.50 | 8.90 | 8.60 | 7.28 |
| | GPT4 | 9.00 | 8.90 | 6.10 | 7.10 | 6.20 | 9.30 | 9.30 | 9.20 | 8.14 |
| | Ours | 8.50 | 7.50 | 6.40 | 3.10 | 4.40 | 5.80 | 9.60 | 9.60 | 6.86 |
| **Second Turn** | PolyLM | 1.30 | 1.00 | 1.50 | 1.10 | 1.00 | 1.20 | 1.00 | 1.10 | 1.15 |
| | X-Llama | 2.60 | 3.60 | 2.50 | 1.20 | 1.80 | 1.70 | 3.20 | 2.90 | 2.44 |
| | Typhoon | 3.00 | 5.20 | 4.10 | 1.70 | 2.70 | 1.80 | 5.90 | 4.80 | 3.65 |
| | PLUG | 2.20 | 2.60 | 1.40 | 0.50 | 2.10 | 1.30 | 2.90 | 3.90 | 2.11 |
| | NLLB-bridge | 5.30 | 4.20 | 4.10 | 2.80 | 2.30 | 3.50 | 4.20 | 6.30 | 4.09 |
| | Llama-2-Chat (EN) | 6.80 | 7.10 | 4.20 | 3.70 | 3.30 | 3.80 | 7.30 | 9.70 | 5.74 |
| | ChatGPT | 3.50 | 7.90 | 5.20 | 3.50 | 5.10 | 7.20 | 6.70 | 8.80 | 5.99 |
| | GPT4 | 8.30 | 8.50 | 4.70 | 4.80 | 7.00 | 8.80 | 8.00 | 8.60 | 7.34 |
| | Ours | 7.50 | 7.30 | 5.60 | 2.10 | 5.20 | 4.80 | 8.20 | 8.70 | 6.18 |

Table 19: GPT-4 evaluation scores on MT-Bench in for different models.

| Model | Helpful-Base | Koala | Oasst | Self-Instruct | Vicuna | All |
|---|---|---|---|---|---|---|
| X-Llama | 2.80 | 3.86 | 3.95 | 3.90 | 4.80 | 3.82 |
| PLUG | 4.88 | 5.47 | 5.23 | 5.32 | 6.90 | 5.41 |
| NLLB-bridge | 4.36 | 4.97 | 5.04 | 4.49 | 4.78 | 4.72 |
| ChatGPT | 7.39 | 7.32 | 7.49 | 7.77 | 8.06 | 7.59 |
| GPT-4 | 9.53 | 9.17 | 9.19 | 8.90 | 9.44 | 9.18 |
| Ours | 8.72 | 7.91 | 7.87 | 7.61 | 8.71 | 8.02 |

Table 20: GPT-4 evaluation scores on Alpaca-Eval in TH for different models.

| Model | First Turn | Second Turn |
|---|---|---|
| Ours | 6.86 | 6.18 |
| w/ base model | 5.56 | 3.08 |
| w/o TH pre-train | 5.55 | 4.44 |
| w/o translation pre-train | 6.55 | 5.04 |
| w/ GPT-4 KD | 5.96 | 4.68 |
| w/o LoRA | 4.58 | 3.34 |
| w/ TH history | - | 5.43 |

Table 21: GPT-4 evaluation scores for ablation studies on MT-Bench in TH.

## B.2 RESULTS IN SCORES

We provide evaluation scores when transforming Llama-2 to TH in Table 18, 19, 20 , and 21.

## B.3 ATTENTION MAP OF THE TRANSLLM OUTPUT

As shown in Figure 4, the TH response focuses on the TH response, EN response, EN query, and TH query, in order from high to low.

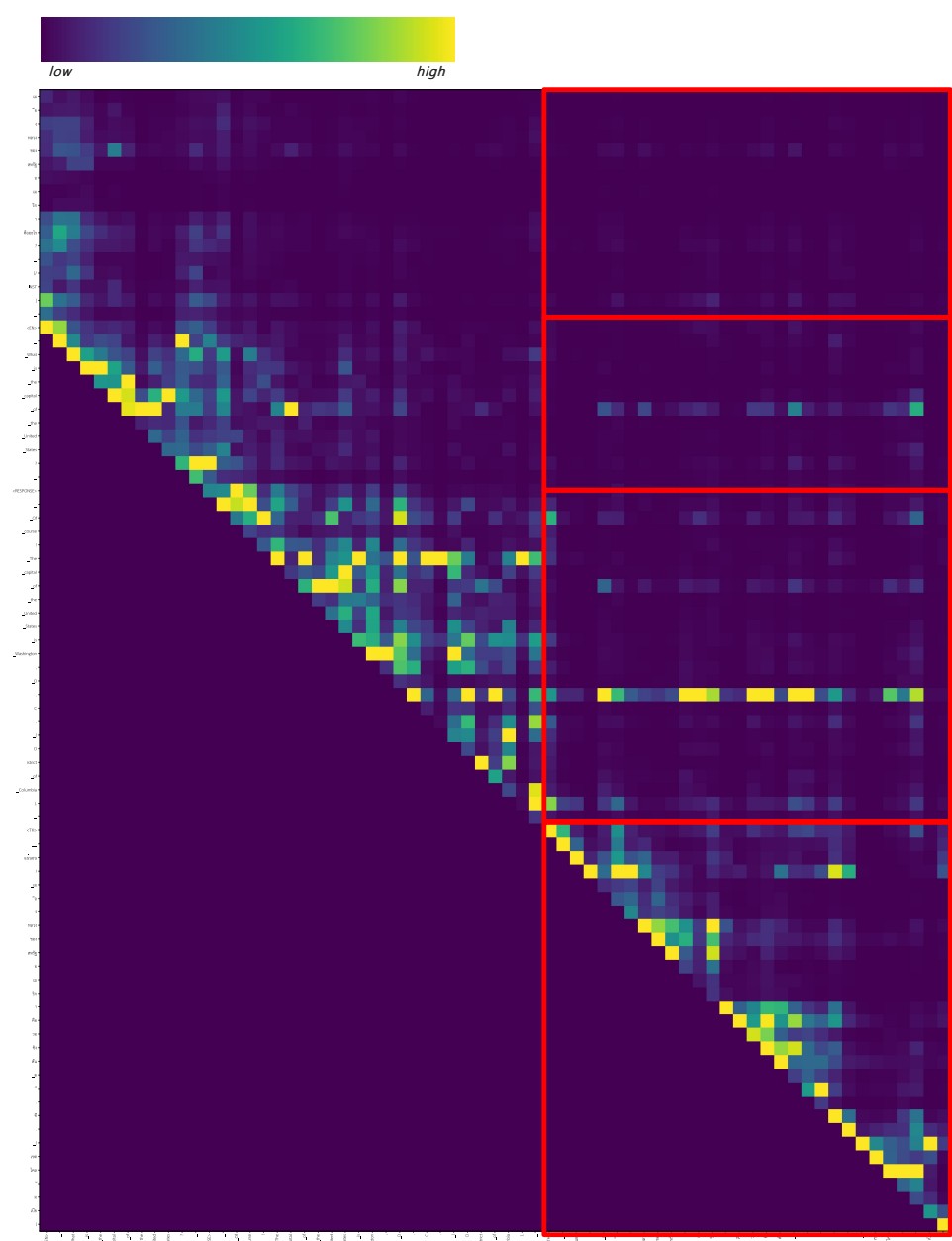

Figure 4: Attention map of the TransLLM output. We mark the attention scores of TH responses with red rectangles. Rectangles from top to bottom indicate attention scores of TH response for TH query, EN query, EN response, and TH response respectively.

## C STATISTICAL METHODS

### C.1 SIGNIFICANT TEST

We conduct a two-sided binomial test for the win rate without tie $p_{\text{win}} = n_{\text{win}}/(n_{\text{win}} + n_{\text{loss}})$. The null hypothesis is that the win rate is not different from the loss rate, i.e. $H_0 : p_{\text{win}} = p_{\text{loss}} = 0.5$, alternative hypothesis $H_1 : p_{\text{win}} \neq 0.5$. We conduct the $\chi^2$ test for safety results in Table 3.

