# OpenReview forum: "Why Not Transform Chat Large Language Models to Non-English?"
_ICLR.cc/2025/Conference — Submitted to ICLR 2025_

### Official Review · Reviewer_T1tU · 2024-11-01

**Soundness:** 2
**Presentation:** 3
**Contribution:** 3
**Rating:** 6
**Confidence:** 3

**Summary:**

This paper focus how to transform a chat LLM from English to non-dominant languages without supervised data. The authors propose a framework named TransLLM,  which incorporate two stage training. In the pretraining stage, the proposed method pretrain the model using monolingual data and parallel translation data to improve the model capabilities on target language. Then, the authors introduce translation COT to transfer advanced chat capabilities from original LLM through SFT, utilizing recovery KD combined with LoRA to mitigate catastrophic forgetting during the transformation.

**Strengths:**

1. **Superior Performance** The experimental results on multiple benchmarks demonstrate that the proposed method outperforms ChatGPT and other strong baselines in both human evaluations and assessments using LLMs as judge.
2. **Solid Experiments** The authors conduct extensive experiments across various languages and LLMs, and validate the effectiveness of each module through tailored ablation studies.

**Weaknesses:**

1. **Lack of a baseline that directly employs multilingual supervised fine-tuning (MSFT)** The authors use google translation to construct TCOT data. Therefore, it is necessary to compare the performance of the baseline that directly conducts MSFT using google-translated SFT data with the proposed method. Compared to this simple baseline, TCOT clearly requires higher inference overhead, so its performance improvements must be significant to be acceptable.
2. **Lack of an ablation experiment to demonstrate the necessity of maintaining the original model parameters until the entire training process is complete**
There are two stages in the entire training process: pretraining and transfer fine-tuning. According to the description in lines 418-423 (Table7 Line4), the authors merged LoRA parameters after the pretraining stage and then performed full-parameter tuning during transfer stage. This ablation experiment only shows that LoRA technologies can mitigate catastrophic forgetting to some extent, which has been discussed in previous work, but it does not prove the necessity of preserving the original LLM parameters throughout the training process. To support this point, the authors should conduct an additional ablation experiment where parameters are merged after the pretraining stage and then fine-tuned using LoRA in transfer stage, comparing it with a version that does not combine parameters until the end of the two-stage training.
3.  I provide some minor concerns and suggestions in the following section.

**Questions:**

# Questions
1. **Lack of implementation details about baseline *NLLB-bridge (Line 353)*** Specifically, how dose NLLB-3B as the bridge between Llama2 and the TH language? In the training stage or the inference stage? Does it utilize TCOT, or does it simply translate the response?

# Suggestions
1.  **Consider using the term "original chat LLM" or a similar alternative instead of "target chat LLM"(like in line 185)** While I understand that the authors use this term to refer to the chat LLM they aim to transform, the term "target" is often associated with "target language" in multilingual and translation scenarios, which refers to the language into which content is being transformed or translated. This may lead to potential misunderstandings and confusion.
2. **Consider conducting experiments with other open-source LLMs like Mistral and Gemma** Although the authors have conducted extensive experiments on Llama2/3/3.1, adding additional experiments with different series of LLMs could further demonstrate the generalizability of the proposed framework.

---

> ### Author Response · Authors · 2024-11-21
>
> Thanks for your time and constructive comments.
>
> > Weaknesses 1: Lack of a baseline that directly employs multilingual supervised fine-tuning (MSFT)
>
> Thank you for your valuable suggestion. We have conducted MSFT on Llama3.1, and the results are presented in the following tables. Multilingual TransLLM (Llama-3.1 to Multilingual in table 5 and 6) demonstrates significantly superior performance compared to MSFT across five languages.
>
> Comparison between multilingual TransLLM and MSFT on MT-bench:
>
> | Lang   | Turn   | Win (%) | Tie (%) | Loss (%) | Δ |
> |--------|--------|---------|---------|----------|----------|
> | TH  | First  | 56.25   | 18.75   | 25.00    | 31.25 |
> | TH  | Second | 63.75   | 16.25   | 20.00    | 43.75 |
> | AR  | First  | 56.25   | 20.00   | 23.75    | 32.50 |
> | AR  | Second | 57.50   | 20.00   | 22.50    | 35.00 |
>
> Comparison between multilingual TransLLM and MSFT on Aya-dataset:
>
> | Lang   | Win (%) | Tie (%) | Loss (%) | Δ |
> |--------|---------|---------|----------|----------|
> | PT  |  53.20   | 23.20   | 23.60    | 29.60 |
> | TE  |  62.80   | 10.80   | 26.40    | 36.40 |
> | TR  |  70.00   | 9.60  | 20.40    | 49.60 |
>
> Furthermore, we would like to emphasize that similar evidence has been presented in previous results.  In previous experiments, we included supervised fine-tuning baselines for a single language (SSFT). As demonstrated in both prior work (Zhu et al., 2023) and this study, MSFT only achieves performance comparable to SSFT. More importantly, we incorporated state-of-the-art open-source baselines such as Typhoon for TH and AceGPT for AR, and TransLLM significantly outperforms these baselines.
>
> > Weaknesses 2: Lack of an ablation experiment to demonstrate the necessity of maintaining the original model parameters until the entire training process is complete
>
> The absence of such an ablation model has been addressed. We conduct the experiment and present the results in the table below. When using LoRA during fine-tuning but not during pre-training, there is also a noticeable performance degradation. However, it still outperforms TransLLM without LoRA throughout the training process. This underscores the importance of preserving the original model parameters until the training process is fully completed.
>
> Comparison between TransLLM and the ablation model w/o LoRA pre-train:
>
> | Turn   | Win (%) | Tie (%) | Loss (%) | Δ |
> |--------|---------|---------|----------|----------|
> | First  | 63.75   | 13.75   | 22.50    | 41.25 |
> | Second | 66.25   | 13.75   | 20.00    | 46.25 |
>
> > Question 1: Lack of implementation details about baseline NLLB-bridge
>
> We use NLLB-3B as a bridge during the inference stage by simply translating the query and response. Specifically, given a query $q^{\beta}$ in target langugae $\beta$, the NLLB-3B model first translates it into English as $q^{EN}$. The original chat LLM (LLaMA2) then generates an English response $a^{EN}$ based on $q^{EN}$. Finally, $a^{EN}$ is translated back into the target language $\beta$ using NLLB-3B, producing the final response $a^{\beta}$. These details will be included in the paper.
>
> > Suggestion 1: Consider using the term "original chat LLM" or a similar alternative instead of "target chat LLM"
>
> Thanks very much for your constructive suggestion. We will change the "target chat LLM" to "original chat LLM" carefully.
>
> > Suggestion 2: Consider conducting experiments with other open-source LLMs like Mistral and Gemma
>
> Thank you for your suggestion. Unfortunately, due to resource constraints, particularly for pre-training, we may be unable to conduct these experiments at this time. However, we plan to explore experiments with other, potentially more powerful, open-source LLMs in the future. Additionally, LLaMA-3 introduces significant advancements in training, data, and architecture, making it substantially different from its predecessor, LLaMA-2. As such, the experiments we have conducted already demonstrate the generalizability of the proposed framework.
>
> **References**
>
> Zhu et al., 2023. Extrapolating large language models to non-english by aligning languages.

---

> ### Author Response · Authors · 2024-11-25
> **Looking forward to your response!**
>
> Thanks for your comments again. We would like to know if our response addressed your concerns. We look forward to discussing these concerns with you.

---

> > ### Comment · Reviewer_T1tU · 2024-11-25
> > **Response to Rebuttal**
> >
> > Thank you for your clarification and the additional experiments, which have addressed some of my concerns. However, I am still unclear about your details regarding the missing ablation experiment. Could you provide more details about your implementation? Specifically, I would like to know whether you perform continual pretraining using LoRA, and if so, whether you merge the parameters before the second training step of transfer tuning with LoRA (Compared with TransLLM). Based on your current description of "w/o LoRA pre-train," it seems that you may not be conducting any continual pretraining at all.

---

> ### Author Response · Authors · 2024-11-25
> **The Implementation of "w/o LoRA pre-train"**
>
> Thank you for your response! We conducted the ablation experiment "w/o LoRA pre-train" in strict accordance with your suggestion. Specifically, we conduct continual pre-training with LoRA and merge the parameters afterward to approximate full-parameter pre-training. Subsequently, LoRA is applied for fine-tuning during the transfer stage. The remaining settings are consistent with those used in the ablation experiments in Table 7. For clarity, we we would like to rename this experiment to "w/o LoRA in pre-training". We sincerely thank you for your valuable contributions to improving our paper!

---

> > ### Comment · Reviewer_T1tU · 2024-11-25
> > **Response to Authors**
> >
> > Thank you for the detailed supplementary experiments.  I will increase my rating to 6.

---

> > > ### Author Response · Authors · 2024-11-25
> > > **Thanks again！**
> > >
> > > Thank you for increasing the score. I would sincerely appreciate it if you could further consider raising the score, taking into account the contribution of our work. If you have any further concerns, we would be happy to address them.

---

### Official Review · Reviewer_pYCd · 2024-11-04

**Soundness:** 3
**Presentation:** 3
**Contribution:** 3
**Rating:** 6
**Confidence:** 3

**Summary:**

The paper introduces two strategies for transforming existing English models to include more languages: 1) low-rank adaptation and 2) recovery knowledge distillation. By experimenting with Llama2/3-8b, they show models have good performances on translated version of AlpacaEval, xCOPA and AdvBench.

**Strengths:**

The strategies introduced in this paper provide a helpful guide on how to transform a pretrained English-centric chat model to include an additional language by leveraging existing multilingual data, additional translation training, LoRA and knowledge distillation to obtain good performances on standardized benchmarks.

**Weaknesses:**

- The evaluation benchmark is mainly obtained from translation. The training strategies involved decide that the chat models will mainly rely on their reasoning and generation capabilities in English, this is even further enhanced by training on translation tasks, therefore they can consequently perform well on the translated benchmarks. I wonder if the authors have tested on non-translated benchmarks?
- The evaluation is done by comparing with GPT4/ChatGPT via using GPT4 as a judge despite the authors mentioning that GPT4 and human annotators exhibit high consistency in the rating process. I wonder if there are accuracy measures for evaluation datasets like XCOPA instead of comparison with the GPT4 to more accurately observe the model’s performance. It would also be helpful to provide example generations to observe models’ responses more qualitatively after adaption for more thorough error analysis.

**Questions:**

Since the strategies mentioned in the paper can effectively reduce catastrophic forgetting. Have the authors tried to transforming the chat models to more than one languages at the same time and compare their performances across languages? Also does the model still maintain high performance in English after the transformation?

---

> ### Author Response · Authors · 2024-11-21
> **Response to Weaknesses 1-2**
>
> Thanks for your time and constructive comments.
>
> > Weaknesses 1: The evaluation benchmark is mainly obtained from translation. The training strategies involved decide that the chat models will mainly rely on their reasoning and generation capabilities in English, this is even further enhanced by training on translation tasks, therefore they can consequently perform well on the translated benchmarks. I wonder if the authors have tested on non-translated benchmarks?
>
> Yes, we had evaluated our method on non-translated benchmarks. Specifically, the human-annotated test set of the Aya-Dataset used in Table 6 serves as a non-translated benchmark. Singh et al. (2024) encourage annotators to contribute fresh samples representative of their native language. We would like to discuss this point to further improve the significance of our method.
>
> For safety, we also engaged professional translators to test a small set of queries in TH and AR (fewer than 10 queries) designed to prompt TransLLM into generating taboo content related to local customs, politics, and similar sensitive topics. In all cases, TransLLM successfully refused to generate harmful content.
>
> Additionally, constructing multilingual benchmarks by translating English datasets is a widely adopted practice due to resource constraints. For example, XCOPA (Ponti et al., 2020) and the dolly-human-edited test set of the Aya-Dataset (Singh et al., 2024) follow this approach. Similarly, we applied this methodology to MT-Bench and Alpaca-Eval, which encompass diverse domains. To ensure high-quality translations, we collaborated with professional translators, aligning with the standards set by prior works.
>
> > Weaknesses 2: The evaluation is done by comparing with GPT4/ChatGPT via using GPT4 as a judge despite the authors mentioning that GPT4 and human annotators exhibit high consistency in the rating process. I wonder if there are accuracy measures for evaluation datasets like XCOPA instead of comparison with the GPT4 to more accurately observe the model’s performance. It would also be helpful to provide example generations to observe models’ responses more qualitatively after adaption for more thorough error analysis.
>
> We appreciate your constructive feedback and would like to clarify a few points regarding our evaluation methodology.
>
> In our paper, we evaluated TransLLM across various dimensions, including instruction following, multi-turn conversation, commonsense reasoning, and safety. Specifically for commonsense reasoning and safety, we provided accuracy measures on benchmark datasets such as XCOPA (Table 17) and AdvBenchMark (Table 3).
>
> For multi-turn conversation, labeling outputs as strictly correct or incorrect may fail to capture nuances and subtle differences in conversational quality. Consequently, comparative evaluations between model outputs have become a widely accepted practice in this domain (e.g., Zheng et al., 2024; Üstün et al., 2024). To provide a more discriminative analysis, we included evaluation scores (on a 1-10 scale) judged by GPT-4, as shown in Tables 18-21.
>
> Additionally, we acknowledge the value of qualitative insights and case studies. As mentioned in Section 4.3 of our paper, we analyzed error cases and attributed most of the observed issues to limitations in English capabilities during the TCOT process. We would be happy to include further example generations to enhance qualitative observations and facilitate a more detailed error analysis.
> We hope these clarifications address your concerns regarding the robustness of our evaluation approach.

---

> > ### Comment · Reviewer_pYCd · 2024-11-26
> >
> > Regarding non-translated benchmarks, I understand that constructing multilingual benchmarks from English is standard practice. However, it would be great if the authors could test on more non-translated benchmarks to show the robustness of the model performance. For example, have the authors considered testing on the Thai subset of M3Exam dataset?

---

> > > ### Author Response · Authors · 2024-11-26
> > > **The M3Exam results will be available soon!**
> > >
> > > We appreciate your constructive suggestions. M3Exam is a valuable multilingual (as well as multimodal and multilevel) benchmark, based on real, official human exam questions. Testing on M3Exam will further enhance the significance of our paper. We plan to test the Thai subset of M3Exam and will do our best to provide the results before the extended deadline.
> > >
> > > However, we would like to highlight that we have already tested on several non-translated benchmarks and observed significant performance improvements. We hope that our responses have addressed all of your concerns.

---

> > > ### Author Response · Authors · 2024-11-28
> > > **The M3Exam results**
> > >
> > > We follow the zero-shot setting from (Zhang et al., 2024) to evaluate various models on the M3Exam dataset (https://github.com/DAMO-NLP-SG/M3Exam). The results are provided in the following table. TransLLM (Llama3.1 to Multilingual in Tables 5 and 6) significantly improves the original Thai performance and outperforms ChatGPT by a substantial margin. Although TransLLM still lags behind GPT-4, we will show later that this is primarily due to the limitations of the original English capabilities.
> > >
> > > Accuracy of various models in Thai on the M3Exam. *The results for ChatGPT and GPT-4 are from Zhang et al. (2024):
> > >
> > > | Model | Acc (%) |
> > > |--------|---------|
> > > | random  | 25.00   |
> > > | Llama 3.1 | 28.09  |
> > > | ChatGPT * | 34.09  |
> > > | TransLLM | 46.68 |
> > > | GPT-4 * | 56.04  |
> > >
> > > Through case analysis, we observed that TransLLM can effectively answer social questions related to local information. An example is provided below:
> > > > "นักท่องเที่ยวคณะหนึ่ง เดินทางท่องเที่ยวทางเรือ ประทับใจกับภูมิประเทศที่สวยงาม เช่น ถ้ำลอด เขาพิงกัน เขาตะปู และยังได้ชมซากดึกดำบรรพ์ที่สุสานหอย" นักท่องเที่ยวคณะนี้ เดินทางท่องเที่ยวบริเวณใดของประเทศไทย
> > >
> > > > The question translated using Google Translate (for presentation purposes only): "A group of tourists traveled by boat and were impressed by the beautiful landscapes such as Tham Lod, Khao Phing Kan, Khao Tapu, and also saw fossils at the Shell Cemetery." Which area of Thailand did this group of tourists travel to?
> > >
> > > TransLLM correctly answers this question with "ชายฝั่งทะเลอันดามัน" (Andaman Coast). However, it often struggles with challenging questions due to the original English capabilities of Llama 3.1.
> > >
> > > To further verify this, we compare LLaMA 3.1 with ChatGPT and GPT-4 on M3Exam in English. As shown in the table below, LLaMA 3.1's performance in English is even worse than that of ChatGPT.
> > >
> > > Accuracy of various models in English on the M3Exam. *The results for ChatGPT and GPT-4 are from Zhang et al. (2024):
> > >
> > > | Model | Acc (%) |
> > > |--------|---------|
> > > | random  | 25.00   |
> > > | Llama 3.1 | 73.99  |
> > > | ChatGPT * | 75.98  |
> > > | GPT-4 * | 87.55  |
> > >
> > > **References**
> > >
> > > Zhang et al., 2024. M3Exam: A Multilingual, Multimodal, Multilevel Benchmark for Examining Large Language Models.

---

> ### Author Response · Authors · 2024-11-21
> **Response to Question 1 and References**
>
> > Question 1: Since the strategies mentioned in the paper can effectively reduce catastrophic forgetting. Have the authors tried to transforming the chat models to more than one languages at the same time and compare their performances across languages? Also does the model still maintain high performance in English after the transformation?
>
> We tried to transform Llama-3.1-instruct to five langagues at the same time, with the results presented in Tables 5 and 6. TransLLM demonstrates competitive performance across languages. The model still maintains high performance in English after the transformation, as illustrated in the subsequent table. The performance of the multilingual TransLLM in English is comparable to that of the original LLM (Llama-3.1), with no statistically significant difference.
>
> Comparison between multilingual TransLLM and Llama-3.1-instruct on MT-bench in English:
>
> | Turn   | Win (%) | Tie (%) | Loss (%) | Δ |
> |--------|---------|---------|----------|----------|
> | First  | 13.75   | 65.00   | 21.25    | -7.50 |
> | Second | 16.25   | 60.00   | 23.75    | -7.50 |
>
> **References**
>
> Singh et al., 2024. Aya dataset: An open-access collection for multilingual instruction tuning
>
> Ponti et al., 2020. XCOPA: A multilingual dataset for causal commonsense reasoning.
>
> Zheng et al., 2024. Judging llm-as-a-judge with mt-bench and chatbot arena.
>
> Üstün et al., 2024. Aya Model: An Instruction Finetuned Open-Access Multilingual Language Model.

---

> ### Author Response · Authors · 2024-11-25
> **Looking forward to your response!**
>
> Thanks for your comments again. We would like to know if our response addressed your concerns. We look forward to discussing these concerns with you.

---

> ### Comment · Reviewer_pYCd · 2024-11-29
>
> Thank you for providing the updated the results promptly! I'll raise my score to 6 and also contribution and presentation scores.

---

> > ### Author Response · Authors · 2024-11-29
> > **Thanks again!**
> >
> > Thank you for raising the score. If you have any further concerns, we would be happy to address them.

---

### Official Review · Reviewer_GsyR · 2024-11-04

**Soundness:** 3
**Presentation:** 3
**Contribution:** 2
**Rating:** 5
**Confidence:** 4

**Summary:**

The paper introduces TransLLM, a framework for adapting large language models (LLMs) that is predominantly trained on English data to other langages.

TransLLM use low-rank adaptation (LoRA) to train monolingual LLMs through three steps: (1) target language pre-training, (2) translation pre-training, and (3) transfer fine-tuning.
The last step uses multiple data format including recovery knowledge distillation (KD) data, translated chain-of-thought (TCOT) data, and bi-direction translation to update LoRA parameters.

The approach is evaluated across five languages and different benchmarks, demonstrating improvements over baselines, surpassing GPT-4 in Thai for helpfulness and safety.

**Strengths:**

+ The paper addresses an important challenge in adapting LLMs to non-English languages, which is crucial for addressing the multilingual generalization problem of LLMs.

**Weaknesses:**

+ The paper is an engineering-focused LLMs training project without introducing any novelty or new methodology for the target problem.
+ The paper is poorly written with contextually unclear terminologies (chat LLMs, Knowledge Distillation, Chain-of-thought).
+ The paper focus on GPT-4 as the main baseline, while GPT-4 multilingual capacity is not properly documented. A multilingual LLM baseline would be more appropriate.
+ The main evaluation is heavily focused on Thai, with limited results for other languages.

**Questions:**

+ Can the proposed method applied to any LLM instead of chat LLM in particular?
+ Is there any distillation of knowledge involved in the training process?
+ Can you elaborate on the "chain-of-thought" in TCOT data format?

---

> ### Author Response · Authors · 2024-11-21
> **Response to Weaknesses 1-2**
>
> Thanks for your time and comments.
>
> > Weaknesses 2: The paper is poorly written with contextually unclear terminologies (chat LLMs, Knowledge Distillation, Chain-of-thought).
>
> We would like to address this concern first to ensure clarity in subsequent discussions.
>
> Chat LLMs: We introduced the terms base LLMs and chat LLMs in lines 41 and 46. Base LLMs refer to foundational models pre-trained on vast corpora using next-word prediction objectives, whereas chat LLMs are derived from these base LLMs through additional post-training techniques (e.g., supervised fine-tuning and reinforcement learning from human feedback) that enable them to follow instructions and align with human preferences.
>
> Knowledge Distillation: We introduced the concept of knowledge distillation for LLMs in lines 42 and 181. A practical approach to enabling base LLMs to follow human instructions effectively is to distill instruction-following knowledge from powerful closed-source LLMs, such as GPT-4. For example, the widely-used Alpaca dataset (Taori et al., 2023) generates queries through the self-instruct technique (Wang et al., 2022) and collects corresponding responses from ChatGPT or GPT-4. While traditional knowledge distillation (Hinton et al., 2015) focuses on learning a teacher's output distribution, knowledge distillation for LLMs, as highlighted by West et al. (2021) and Wang et al. (2022), instead relies on distilling knowledge from sampled data, since the full distribution of closed-source LLMs is typically inaccessible.
>
> Chain-of-thought: We introduced the concept of translation chain-of-thought (TCOT) in lines 74 and 193. TCOT performs the response generation in the target language as a sequence of sub-tasks step by step: It first translates the query to English; then responds to the query in English; and finally generates the non-English answer based on all the above context.
>
> > Weaknesses 1: The paper is an engineering-focused LLMs training project without introducing any novelty or new methodology for the target problem.
>
> Significant practical impact: The proposed TransLLM framework addresses a critical gap in transforming chat LLMs to target languages, which has not been effectively tackled in prior work.
>
> Innovative integration and novel techniques: To enable efficient transfer, TransLLM employs TCOT to decompose the transfer process into common sub-tasks, eliminating the need for specific training data. More importantly, TransLLM improves sub-task performance through continual pre-training. To mitigate catastrophic forgetting, we propose novel the Recovery KD combined with LoRA finetuning. While components like TCOT and LoRA are established methods, their integration within the TransLLM framework is highly innovative. As noted by reviewers vJuZ and T1tU, our ablation studies demonstrate that TCOT and LoRA alone are insufficient without the full TransLLM framework.
>
> Novel findings and insights:  Our study highlights that relying on widely used GPT-4 KD can be misleading when transforming chat LLMs. Additionally, our analysis (Section 4.2) provides valuable insights into how Recovery KD + LoRA effectively mitigates catastrophic forgetting.
>
> Superior performance and resource efficiency: Experimental results confirm the effectiveness, flexibility, and generalizability of TransLLM across diverse languages and LLMs. The framework outperforms even GPT-4 in Thai with an 8B model, showcasing its effectiveness and demonstrating a practical pathway to achieving similar results with significantly fewer resources.

---

> ### Author Response · Authors · 2024-11-21
> **Response to Weaknesses 3-4**
>
> > Weaknesses 3: The paper focus on GPT-4 as the main baseline, while GPT-4 multilingual capacity is not properly documented. A multilingual LLM baseline would be more appropriate.
>
> Thank you for your valuable suggestions. As demonstrated in both prior work (Zhu et al., 2023) and this study, multilingual LLMs generally perform on par with language-specific LLMs. Beyond GPT-4, our study already incorporates strong baselines tailored to individual languages, such as Typhoon for Thai (TH) and AceGPT for Arabic (AR), which are the state-of-the-art open-source LLMs for these languages. Therefore, our baselines already provide sufficient validation. GPT-4 serves as a valid baseline, given its well-documented multilingual capabilities as outlined in its technical reports and those of Llama-3, both of which were cited in the previous version of the paper. For a more direct comparison, we have added multilingual supervised fine-tuning (MSFT) on Llama3.1 as a baseline, with results presented in the following tables. Multilingual TransLLM (Llama-3.1 to Multilingual in table 5 and 6) demonstrates significantly superior performance compared to MSFT across five languages.
>
> Comparison between multilingual TransLLM and MSFT on MT-bench:
>
> | Lang   | Turn   | Win (%) | Tie (%) | Loss (%) | Δ |
> |--------|--------|---------|---------|----------|----------|
> | TH  | First  | 56.25   | 18.75   | 25.00    | 31.25 |
> | TH  | Second | 63.75   | 16.25   | 20.00    | 43.75 |
> | AR  | First  | 56.25   | 20.00   | 23.75    | 32.50 |
> | AR  | Second | 57.50   | 20.00   | 22.50    | 35.00 |
>
> Comparison between multilingual TransLLM and MSFT on Aya-dataset:
>
> | Lang   | Win (%) | Tie (%) | Loss (%) | Δ |
> |--------|---------|---------|----------|----------|
> | PT  |  53.20   | 23.20   | 23.60    | 29.60 |
> | TE  |  62.80   | 10.80   | 26.40    | 36.40 |
> | TR  |  70.00   | 9.60  | 20.40    | 49.60 |
>
> > Weaknesses 4: The main evaluation is heavily focused on Thai, with limited results for other languages.
>
> Our evaluation approach aligns with common practices in related work. For instance, Chat Vector (ACL 2024) primarily focuses on Chinese and extends its evaluation to Korean to demonstrate versatility. To thoroughly evaluate our method with limited resources, we adopt a twofold strategy: an in-depth assessment of a single language and a broad evaluation across multiple languages. Specifically, we comprehensively evaluate TransLLM in Thai (TH) across various aspects, including instruction following, multi-turn conversation, commonsense reasoning, and safety. Additionally, we extend the evaluation of instruction-following capabilities to four other languages.

---

> ### Author Response · Authors · 2024-11-21
> **Response to Question 1-3 and References**
>
> > Question 1: Can the proposed method applied to any LLM instead of chat LLM in particular?
>
> Although Table 7 shows that transforming chat LLMs achieves better performance than base LLMs, our method is not limited to chat LLMs and can also be applied to base LLMs, as demonstrated in the following table.
>
> Comparison between TransLLM w/ base LLMs and X-Llama:
>
> | vs. Model | Turn   | Win (%) | Tie (%) | Loss (%) | Δ |
> |--------|--------|---------|---------|----------|----------|
> | X-Llama | First  | 58.75   | 18.75   | 22.50    | 36.25 |
> | X-Llama | Second | 33.75   | 51.25   | 15.00    | 18.75 |
>
> If "any LLM" refers to domain-specific LLMs such as code LLMs, json LLMs, or math LLMs, it is worth noting that the MT-Bench used in our evaluation already encompasses these domains. Therefore, TransLLM is expected to perform effectively across them.
>
> > Question 2: Is there any distillation of knowledge involved in the training process?
>
> As demonstrated in Section 4.2, transfer training causes the LLM to significantly forget its original English knowledge (Table 8). The proposed recovery knowledge distillation method distills the original English knowledge to the trained LLM. This approach encourages the LLM to learn a generalizable pattern that minimizes the influence of LoRA parameters when generating English content, thereby restoring its original English knowledge (line 457).
>
> > Question 3: Can you elaborate on the "chain-of-thought" in TCOT data format?
>
> Chain-of-thought (Wei et al., 2022) breaks complex problems into a series (chain) of manageable steps (thoughts). In TCOT, these steps are pre-defined, drawing inspiration from how humans handle a second language. As illustrated in line 194, solving the query $q^{TH}$ in the target language TH using TCOT involves three steps: first, translating the query into English ($q^{EN}$); second, answering the query in English ($a^{EN}$); and finally, generating the answer in the target language ($a^{{TH}}$) based on the preceding steps. Special tokens are employed to guide the behavior of each step. Accordingly, the input of TCOT data is $x = q^{TH}$, the desired output is $y = \text{cat}(\langle EN \rangle, q^{EN},\langle\text{RESPONSE}\rangle, a^{EN},\langle TH \rangle, a^{{TH}})$, where $\langle\cdot\rangle$ denotes the special token defined in paper.
>
> **References**
>
> Taori et al., 2023. Stanford alpaca: an instruction-following llama model.
>
> Wang et al., 2022. Self-instruct: Aligning language models with self-generated instructions.
>
> Hinton et al., 2015. Distilling the knowledge in a neural network.
>
> West et al., 2021. Symbolic knowledge distillation: from general language models to commonsense models.
>
> Wei et al., 2022. Chain-of-Thought Prompting Elicits Reasoning in Large Language Models.
>
> Zhang et al., 2023. Plug: Leveraging pivot language in cross-lingual instruction tuning
>
> Zhu et al., 2023. Extrapolating large language models to non-english by aligning languages.

---

> ### Author Response · Authors · 2024-11-25
> **Looking forward to your response!**
>
> Thanks for your comments again. We would like to know if our response addressed your concerns. We look forward to discussing these concerns with you.

---

> > ### Comment · Reviewer_GsyR · 2024-11-26
> >
> > Thank you for your detailed response to my comments and for addressing my queries.
> >
> > However, I remain unconvinced about the novelty of the proposed framework. The paper's main innovation appears to be gathering and translating data from different sources for LoRA fine-tuning. The use of LoRA fine-tuning for transferring models to target languages or domains is a well-established technique in the field. Similarly, multi-step continual training is not new; while the idea of finding the appropriate combination of different data types for each corresponding step is somewhat novel, it is not significantly.
> >
> > Regarding GPT-4, its technical report only performs a singular multilingual benchmark (multilingual MMLU) without providing details on how the model was trained for multilingual capabilities. The paper should focus on baselines that emphasize multilinguality (e.g., LLaMA 3.1, Aya-model, BLOOMz, etc) and demonstrate that the targeted approach of TransLLM is able to outperform these generally trained multilingual models to validate its effectiveness.
> > Additionally, The paper should demonstrate the generality and applicability of TransLLM by providing results for diverse languages with varying levels of resource availability.
> >
> > Based on the authors' response, I would like to raise my score for the paper to **4**, acknowledging that some of my concerns regarding the paper's experimental results have been partially addressed. However, I remain convinced that the paper requires further revision before acceptance, particularly in terms of its novelty and experimental focus.

---

> ### Author Response · Authors · 2024-11-26
>
> Thank you for your valuable feedback.
>
> **Novelty.** We would like to further clarify the novelty our work in the context of your concerns. While techniques like LoRA and multi-step continual pre-training are indeed well-established, our work introduces **novel integrations** that distinguish it from existing approaches. As noted by Reviewer vJuZ, our framework offers **"detailed insights into critical decisions"** through ablation studies and in-depth analysis. Each technique we utilize, including TCOT and LoRA, is **motivated by the specific challenges** inherent in transferring chat LLMs. For instance, TCOT enables the transfer of models without requiring advanced supervised data, such as human preference data. We enhance TCOT's performance by incorporating pre-training on translation and target language modeling. One of the **key innovations** in our approach is the **Recovery KD** technique, which, to the best of our knowledge, has not been proposed in prior work. Recovery KD allows us to preserve the original knowledge with LoRA fine-tuning.
>
> Furthermore, our **empirical findings are novel** in demonstrating that the commonly used GPT-4 KD ( i.e. Alpaca dataset) is misleading when applied to chat LLM transformation. We demonstrate that Recovery KD combined with LoRA effectively recovers knowledge by enabling the LLM to learn generalizable patterns, which reduces the impact of LoRA parameters when generating English content. This insight has significant implications for mitigating catastrophic forgetting and could be applied to other tasks as well.
>
> While novelty is one aspect of our contribution, our work also addresses a practical and significant challenge, **as highlighted in your Strengths section**. Our experiments demonstrate that we achieve superior performance despite using limited resources. More importantly, it contributes to enhancing the safety of non-English LLMs.
>
> **Multilingual Baselines.** As you suggested, Llama 3.1 serves as a solid baseline for multilinguality. In our previous experiments, we compared Llama 3.1 on TH in Table 5, demonstrating that TransLLM improves the original TH performance. In Response to Weaknesses 3-4, we further enhance Llama 3.1's multilingual capabilities with MSFT. However, TransLLM still significantly outperforms MSFT across languages. Generally, multilingual LLMs perform on par with, or even worse than, language-specific models. In our previous experiments, we have shown superior performance against strong baselines tailored to individual languages. To further support this, we compare the **Aya model** (https://huggingface.co/CohereForAI/aya-101) on the Aya dataset in PT, TR, and TE, as shown in the table below. TransLLM outperforms the Aya model by a substantial margin.
>
> Comparison between multilingual TransLLM and Aya-model on Aya-dataset:
> | Lang   | Win (%) | Tie (%) | Loss (%) | Δ |
> |--------|---------|---------|----------|----------|
> | PT  |  74.80   | 10.80   | 14.40    | 60.40 |
> | TE  |  66.80   | 9.20   | 24.00    | 42.80 |
> | TR  |  74.00   | 10.00  | 16.00    | 58.00 |
>
> **Language Resource Variation.** Regarding the diversity of language resources, our experiments indeed cover a broad spectrum of resource availability. Following the categorization used in the Aya dataset (Table 5), we included languages such as Arabic (high resource), Thai, Portuguese, and Turkish (medium resource), and Telugu (low resource).
>
> We hope that these clarifications will address your concerns.

---

> > ### Author Response · Authors · 2024-11-26
> > **Thank You for Your Constructive Feedback and Revised Ratings**
> >
> > We are grateful for your reassessment, which led to an increase in the soundness rating to 3, contribution rating to 2, and overall rating to 5. We hope that we have successfully addressed most of your concerns in the rebuttal. If there are any additional concerns or suggestions you would like to discuss, we would be more than happy to address them.

---

### Official Review · Reviewer_vJuZ · 2024-11-05

**Soundness:** 3
**Presentation:** 3
**Contribution:** 3
**Rating:** 8
**Confidence:** 4

**Summary:**

This work proposes a method to equip Chat LLMs with target language capabilities, such that the chat capability in English can be transferred to target languages that the original LLM is not good at or completely lacking. To achieve this, this work introduces a training scheme, aiming at objectives: 1) learn to use translation in the chain-of-thought process (TCOT), serving as the anchor for reasoning in target languages; 2) preservation of original knowledge to avoid catastrophic forgetting.

Specifically, this work builds QA pairs in target languages by translating from English training resources, which are used for TCOT finetuning and translation finetuning. To avoid catastrophic forgetting, the training also includes original QA pairs in English, combined with LoRA finetuning instead of full finetuning, to preserve the original knowledge. Additionally, this work also observes that pretraining on target languages before any finetuning could contribute significantly.

Experiments on conducted on multiple instruction following, reasoning and safety datasets, examining five languages to transfer: Thai, Arabic, Portuguese, Telugu, Turkish. Results show that the transferred LLMs could outperform ChatGPT and other baselines, including Chat Vector (Huang et al., 2024) and translation-as-a-bridge.

**Strengths:**

- The idea to use translation in CoT for target language transfer is intuitive and well-motivated. Experiments show that the overall training scheme is effective, outperforming all baselines except for GPT-4.

- Ablation studies and analysis are performed, providing insights on detailed decisions, e.g. distilling answers from the LLM itself rather than more powerful GPT-4, the importance of target language pretraining, and the effectiveness on preserving original knowledge in the source language.

- Human studies are performed to evaluate the resulting chat quality in target languages.

**Weaknesses:**

Since this work builds new training data by translating from English resources to target languages, it would be important to decouple the improvement from the new data vs. the new training scheme. Several simple baselines could be added:

- Removing TCOT, only performing: 1) target language pretraining; 2) finetuning directly on target language QA pairs. The resulting LLMs may forget original knowledge, but it would be insightful to see the performance on target languages directly.

- For the above process, also add QA pairs in English during finetuning to preserve original knowledge, so to examine how much TCOT really contributes in the transfer process.

The above analysis could increase the significance of this work.

**Questions:**

For the English training resources, is the translation performed for each target languages, with the same amount of translated pairs?

---

> ### Author Response · Authors · 2024-11-21
>
> Thanks for your time and postive comments.
>
> > Weakness 1: Include several baselines to decouple the improvement from the new data vs. the new training scheme.
>
> Thank you for your constructive suggestions. We have added these two baselines following the ablation study settings. As shown in the table below, TransLLM significantly outperforms the baselines, demonstrating the effectiveness of the proposed training scheme. While Recovery KD provides only a slight improvement, combining Recovery KD with LoRA helps recover English knowledge. However, without TCOT, the model struggles to fully leverage this English knowledge.
>
> Comparison between TransLLM and baselines on MT-bench when transforming Llama-2 in Thai:
>
> | Turn   | vs. Model | Win (%) | Tie (%) | Loss (%) | Δ        |
> |--------|-----------|---------|---------|----------|----------|
> | First  | TH pre-train + TH fine-tune   | 65.00   | 15.00   | 20.00    | 45.00 |
> | First  | TH pre-train + TH fine-tune + Recovery KD   |  63.75  | 13.75   | 22.50    | 41.25 |
> | Second | TH pre-train + TH fine-tune   | 66.25   | 15.00   | 18.75    | 47.50 |
> | Second | TH pre-train + TH fine-tune + Recovery KD    | 66.25   | 13.75   | 20.00   | 46.25 |
>
> > Question 1: For the English training resources, is the translation performed for each target language, with the same amount of translated pairs?
>
> Yes, the same Recovery KD dataset is translated into each target language, ensuring that all translated pairs are parallel and have the same quantity.

---

> > ### Comment · Reviewer_vJuZ · 2024-11-25
> >
> > Thanks for the authors' additional experiments.
> >
> > Based on the added experiments, it seems to me that the model trained with TCOT does not completely surpass the model without TCOT, since in 35% cases the model trained with TCOT is not better than the other. It is also unclear the gap between the two models (one might be only always slightly better than the other).
> >
> > To be more convincing, I would further suggest to independently evaluate each model on more available downstream tasks (e.g. QA in English and QA in Thai), and show the gap directly, so to be more clear on the improvement of the proposed TCOT scheme.

---

> > > ### Author Response · Authors · 2024-11-25
> > >
> > > Thank you for your valuable feedback! Regarding the 35% of cases where the model trained with TCOT is not better than the one without TCOT, we believe these results are primarily driven by the inherent difficulty of certain examples, which are either too easy or too challenging to distinguish clearly. For example, GPT-4 only demonstrates an average win rate of only about 60% (compared to GPT-3.5, Claude-V1, Vicuna-13B, Alpaca-13B, and LLaMA-13B) in human evaluations on MT-bench in English, as shown in Zheng et al. (2024). This suggests that the performance gap between models with and without TCOT remains substantial.
> > >
> > > To further clarify the extent of the gap, we provide GPT-4 evaluation scores in the table below, where each model’s response quality is measured on a scale of 1 to 10. More details on the evaluation process can be found in Appendix 4.2 and Zheng et al. (2024). These scores clearly demonstrate that the model trained with TCOT significantly outperforms the ablation baselines, with statistical significance, thereby reinforcing the effectiveness of our proposed method.
> > >
> > >  GPT-4 evaluation scores on MT-Bench in TH for different models:
> > >
> > > | Model | First Turn | Second Turn |
> > > |--------|-----------|---------|
> > > | TH pre-train + TH fine-tune |  4.70  | 3.54   |
> > > | TH pre-train + TH fine-tune + Recovery KD  |  4.76   | 3.88   |
> > > | TransLLM  |  6.86   | 6.18   |
> > >
> > > **Reference**
> > >
> > > Zheng et al., 2024. Judging LLM-as-a-Judge with MT-Bench and Chatbot Arena.

---

### Meta-Review · Area_Chair_8xWa · 2024-12-18

**Metareview:**

The paper presents a method to enhance Chat LLMs with capabilities in target languages that are not well-supported by the original English-centric models. This is achieved through a framework called TransLLM, which employs a strategic training scheme involving several key components. However, the limitations of the paper are 1)  The paper uses GPT-4 as the primary baseline, despite the lack of comprehensive documentation on its multilingual capabilities. It would be more suitable to compare against a multilingual LLM baseline. 2) The main evaluation is heavily focused on Thai, providing limited results for other languages. 3)The proposed training approach offers limited novelty in addressing the target problem. 4) The formatting of the text and tables needs to be improved to make it easier to follow.

**Additional Comments On Reviewer Discussion:**

1) The paper uses GPT-4 as the primary baseline, despite the lack of comprehensive documentation on its multilingual capabilities. It would be more suitable to compare against a multilingual LLM baseline.
2) The main evaluation is heavily focused on Thai, providing limited results for other languages.
3) The proposed training approach offers limited novelty in addressing the target problem.

The authors provided limited positive responses to these issues, leading to my decision to reject the paper.

---

### Decision · Program_Chairs · 2025-01-22

Reject